# Human Mesenchymal Stem Cells Overexpressing Interleukin 2 Can Suppress Proliferation of Neuroblastoma Cells in Co-Culture and Activate Mononuclear Cells In Vitro

**DOI:** 10.3390/bioengineering7020059

**Published:** 2020-06-17

**Authors:** Daria S. Chulpanova, Valeriya V. Solovyeva, Victoria James, Svetlana S. Arkhipova, Marina O. Gomzikova, Ekaterina E. Garanina, Elvira R. Akhmetzyanova, Leysan G. Tazetdinova, Svetlana F. Khaiboullina, Albert A. Rizvanov

**Affiliations:** 1Institute of Fundamental Medicine and Biology, Kazan Federal University, 420008 Kazan, Russia; DaSChulpanova@kpfu.ru (D.S.C.); VaVSoloveva@kpfu.ru (V.V.S.); SSArhipova@kpfu.ru (S.S.A.); MOGomzikova@kpfu.ru (M.O.G.); EEGaranina@kpfu.ru (E.E.G.); ElviRAhmetzyanova@kpfu.ru (E.R.A.); LeGTazetdinova@kpfu.ru (L.G.T.); SFHajbullina@kpfu.ru (S.F.K.); 2Shemyakin-Ovchinnikov Institute of Bioorganic Chemistry, The Russian Academy of Sciences, 117997 Moscow, Russia; 3School of Veterinary Medicine and Science, University of Nottingham, Nottingham LE12 5RD, UK; Victoria.James@nottingham.ac.uk; 4Department of Microbiology and Immunology, University of Nevada, Reno School of Medicine, Reno, NV 89557, USA

**Keywords:** immunotherapy, mesenchymal stem cells, interleukin 2, cancer therapy, neuroblastoma

## Abstract

High-dose recombinant interleukin 2 (IL2) therapy has been shown to be successful in renal cell carcinoma and metastatic melanoma. However, systemic administration of high doses of IL2 can be toxic, causing capillary leakage syndrome and stimulating pro-tumor immune response. One of the strategies to reduce the systemic toxicity of IL2 is the use of mesenchymal stem cells (MSCs) as a vehicle for the targeted delivery of IL2. Human adipose tissue-derived MSCs were transduced with lentivirus encoding *IL2* (hADSCs-IL2) or blue fluorescent protein (BFP) (hADSCs-BFP). The proliferation, immunophenotype, cytokine profile and ultrastructure of hADSCs-IL2 and hADSCs-BFP were determined. The effect of hADSCs on activation of peripheral blood mononuclear cells (PBMCs) and proliferation and viability of SH-SY5Y neuroblastoma cells after co-culture with native hADSCs, hADSCs-BFP or hADSCs-IL2 on plastic and Matrigel was evaluated. Ultrastructure and cytokine production by hADSCs-IL2 showed modest changes in comparison with hADSCs and hADSCs-BFP. Conditioned medium from hADSC-IL2 affected tumor cell proliferation, increasing the proliferation of SH-SY5Y cells and also increasing the number of late-activated T-cells, natural killer (NK) cells, NKT-cells and activated T-killers. Conversely, hADSC-IL2 co-culture led to a decrease in SH-SY5Y proliferation on plastic and Matrigel. These data show that hADSCs-IL2 can reduce SH-SY5Y proliferation and activate PBMCs in vitro. However, IL2-mediated therapeutic effects of hADSCs could be offset by the increased expression of pro-oncogenes, as well as the natural ability of hADSCs to promote the progression of some tumors.

## 1. Introduction

Interleukin 2 (IL2) is one of the first cytokines which, along with interferon α (IFN-α), was used for the treatment of cancer [1]. IL2 is mainly produced by CD4^+^ T-cells and plays a vital role in growth as well as differentiation of T-cells, B cells, natural killer (NK) cells, and many other cell types [2]. It is the ability of IL2 to stimulate the proliferation and activation of immune cells with antitumor activity that made it possible to achieve successes in IL2-based immunotherapy of cancer [3]. High-dose (HD) recombinant IL2 therapy has been shown to be effective and has been approved by the Food and Drug Administration (FDA) for the treatment of renal cell carcinoma and metastatic melanoma [4]. In the case of metastatic renal cell carcinoma, a complete response was observed in 7% of patients, and a partial response was observed in 13% of patients. In the case of metastatic melanoma, 7% of patients achieved a complete response [4]. However, after the therapy with recombinant IL2 was approved by the FDA, it has been found that IL2 regulates not only effector T-cells but also regulatory T-cells (Tregs), which can inhibit an anti-tumor immune response [5,6]. In addition, the use of HD IL2 therapy can lead to increased vascular permeability, hypotension, pulmonary edema, hepatocyte cell death and kidney failure [7]. Changing the way IL2 is administered from single HD to a subcutaneous low-dose has been shown to prevent vascular permeability and hypotension [8]. However, since the half-life of IL2 is a few minutes [9], the use of low-dose therapy does not allow a high concentration of recombinant protein to be maintained during the treatment, which reduces therapeutic efficacy [10]. Interestingly, HD IL2-associated vessel toxicity can be used as a treatment for cancer [11]. Local injection of IL2 in the tumor site induce disruption of tumor vessels only, science tumor blood vessels are more vulnerable than normal blood vessels to the actions of IL2. At the same time, the dose of IL2 used for the local injection is too low to cause side effects [12].

The use of mesenchymal stem cells (MSCs) may be a potential strategy that can reduce systemic toxicity and ensure targeted delivery of IL2 as well as maintain high levels of IL2 in the tumor microenvironment, since MSCs have the ability to migrate toward tumor sites in vivo [13,14,15]. The use of MSCs for the delivery of chemotherapeutics, tumor suppressor proteins, cytokines, oncolytic viruses and microRNAs (miRNAs) has previously been reviewed in detail [16,17,18]. Briefly, genetically modified MSCs overexpressing cytokines/chemokines and other biological factors, have been shown to be effective in activating immune cells and suppressing a number of tumors in vitro and in vivo [19,20,21,22]. For example, MSCs expressing IFN-α were shown to induce cancer cell apoptosis in a metastatic melanoma mouse model [19]. Similarly, genetically modified MSCs expressing tumor necrosis factor (TNF)-related apoptosis-inducing ligand (TRAIL) were shown to induce apoptosis in several tumor types [23,24]. The antitumor activity of MSCs expressing various ILs has also been demonstrated. For example, MSCs expressing IL12 increased NK cell infiltration into tumors formed in a mouse model of glioma [25], as well as decreasing metastasis and inducing cancer cell apoptosis in mice modelling ovarian cancer [26]. In accordance with these findings, a recent study by You et al., showed that amniotic fluid-derived MSCs expressing IL2 can induce ovarian cancer cell apoptosis in vivo [27].

However, development and implementation of MSC-based therapies have several potential drawbacks. MSCs have been reported to support the progression of some tumor types, including head and neck carcinoma [28], glioma [29], and breast cancer [30]. MSC-linked tumor progression is believed to be mediated by growth factors secreted by MSCs, such as vascular endothelial growth factor (VEGF), IL8, transforming growth factor β (TGF-β), epidermal growth factor (EGF) and platelet-derived growth factor (PDGF). In addition, MSCs in the tumor have the potential to differentiate into pericytes and tumor-associated fibroblasts (TAF), thereby promoting a tumorigenic microenvironment [31]. However, contrasting reports support a tumor suppressive role for MSCs in breast cancer [32], Kaposi’s sarcoma [33] and pancreatic cancer [34], potentially through inhibition of the PI3K/AKT pathway and activation of other pathways leading to cell-cycle arrest and apoptosis [35]. These data suggest the effect of MSCs on tumor growth and development depends on various molecules produced by MSCs, however, the mechanisms of double-edged effect of MSCs are yet to be fully understood.

To further address the challenges of using modified MSCs to treat tumors, we generated a human adipose tissue-derived MSCs line (hADSCs) stably expressing IL2, to determine the effects of MSC-mediated IL2 delivery on SH-SY5Y neuroblastoma cell growth and immune modulation in vitro.

## 2. Materials and Methods

### 2.1. Cells and Culture Conditions

Adipose tissue and blood samples were collected from donors at the Republican Clinical Hospital in accordance with approved ethical standards and current legislation (the protocol was approved by the Committee on Biomedical Ethics of Kazan Federal University (No. 3, 03/23/2017)). Informed consent was obtained from each donor.

We isolated hADSCs from two different donors as previously described [36,37]. Briefly, subcutaneous adipose tissue samples were obtained during liposuction procedures. Lipoaspirate was washed (3 × phosphate buffered saline (PBS)) and fermented using 0.2% collagenase (Biolot, St. Petersburg, Russia) at 37 °C for 1 h. Erythrocytes were removed using Red Blood Cell (RBC) Lysis Buffer (BioLegend, San Diego, CA, USA). The viability of the isolated cells was determined using Trypan Blue Solution (0.4%, Gibco, Grand Island, NY, USA) staining. Cells were cultured in Dulbecco’s modified Eagle medium (DMEM)/F12 medium (PanEco, Moscow, Russia) supplemented with 10% fetal bovine serum (FBS, Invitrogen, Waltham, MA, USA), 2 mM L-glutamine and antibiotics (100 U/mL penicillin, 100 μg/mL streptomycin, Biolot, St. Petersburg, Russia) and incubated at 37 °C in a humidified, 5% CO_2_ atmosphere. hADSCs were used up to passages 5–7.

Peripheral blood mononuclear cells (PBMCs) from two different donors were isolated using Ficoll–Paque gradient fractionation as previously described [38]. Cells were cultured in RPMI-1640 medium (PanEco, Moscow, Russia) with 10% FBS (HyClone, Logan, UT, USA), 2 mM L-glutamine and antibiotics (PanEco, Moscow, Russia) and incubated at 37 °C, 5% CO_2_.

Human PC3 prostate cancer cells (ATCC #CRL-1435), SH-SY5Y bone marrow neuroblastoma cells (ATCC #CRL-2266), A549 lung adenocarcinoma cells (ATCC #CCL-185) and 293T embryonic kidney cells (ATCC #CRL-3216) were obtained from the American Type Culture Collection (ATCC, Manassas, VA, USA). Cells were cultured in DMEM/F12 medium (PanEco, Moscow, Russia) with 10% FBS (HyClone, Logan, UT, USA), 2 mM L-glutamine and antibiotics (PanEco, Moscow, Russia) and incubated at 37 °C, 5% CO_2_. Subsequent passage of cells and medium replacement were carried out according to standard protocols. Microscopy was done using an Axio Observer.Z1 (CarlZeiss, Jena, Germany) microscope. Images were analyzed using the software Axio Vision Rel. 4.8.

### 2.2. Differentiation of Mesenchymal Stem Cells (MSCs) into Adipocytes, Chondrocytes and Osteoblasts

Native and genetically modified hADSCs were differentiated into adipogenic, chondrogenic and osteogenic directions using StemPro^®^ Adipogenesis Differentiation Kit (#A10070-01), StemPro^®^ Chondrogenesis Differentiation Kit (#A10071-01) and StemPro^®^ Osteogenesis Differentiation Kit (#A10072-01), respectively (all Gibco, Grand Island, NY, USA) according to the manufacturer’s protocol. For adipogenic differentiation, hADSCs (1 × 10^4^ cells) were seeded in a 24-well plate. When hADSCs reached 70% confluence, culture medium was removed and differentiation medium was added. Differentiation medium was replaced every three days for 14 days. To demonstrate adipogenic differentiation, cells were fixed (10% paraformaldehyde 10 min, room temperature (RT)) and stained with 0.3% oil red O (#O0625, Sigma-Aldrich, St. Louis, MO, USA). For chondrogenic differentiation, hADSCs (5 × 10^4^ cells in 5 μL) were seeded in the center of the well in 24-well plates and incubated for 2 h in a high humidity chamber. At the end of the incubation, chondrogenesis media was added and cells were incubated for 21 days. Chondrogenic differentiation was determined by staining fixed cells (10% paraformaldehyde 10 min, RT) with Alcian blue (1% in 0.1N HCl; #A5268, Sigma-Aldrich, St. Louis, MO, USA). For osteogenic differentiation, hADSCs (1 × 10^4^ cells) were seeded in 24-well plates. When hADSCs reached 70% confluence the culture medium was removed and differentiation medium was added. Differentiation medium was replaced every three days for 28 days. Cells were fixed with 10% paraformaldehyde for 10 min at RT and stained with 1% aqueous AgNO_3_ for 1 h (in the dark) to determine the presence of calcium deposits.

### 2.3. Lentivirus Production

Vector plasmid encoding human *IL2* gene (pLX304-IL2) was obtained from the Harvard Plasmid Database (#HsCD00421565-4). Vector plasmid pLenti CMV green fluorescent protein (GFP) Blast was purchased from Addgene, Watertown, MA, USA (#17445). Vector plasmid pLX303-BFP encoding a blue fluorescent protein (BFP) gene was generated using Gateway cloning (Invitrogen, Waltham, MA, USA). The BFP gene was sub-cloned from the donor vector (pDONR221) into the lentiviral plasmid vector pLX303 by LR recombination using Gateway™ LR Clonase™ II Enzyme mix (#11791020, Invitrogen, Waltham, MA, USA) according to the manufacturer’s instructions.

To produce the second-generation replication-incompetent lentiviruses (LVs), near confluent 293T cells were transfected using calcium phosphate with three plasmids’ encoding: target gene vector; gag/pol genes and additional viral packaging genes (pCMV-dR8.2 dvpr, Addgene #8455, Watertown, MA, USA); and glycoprotein G of the vesicular stomatitis virus gene (pCMV-VSV-G, Addgene #8454, Watertown, MA, USA) [39]. Resulting LV-IL2, LV-BFP and LV-GFP were concentrated by ultracentrifugation (2 h at 26,000 rpm). The viral titer was determined by infecting cells at various dilutions of the viral stock and determining percentage of transduced cells by flow cytometry.

### 2.4. Genetic Modification and Selection

LV-IL2 or LV-BFP were added at a multiplicity of infection (MOI) of 10 to hADSCs (50% confluency) and cells were cultured with the virus in serum-free DMEM/F12 for 6 h. At the end of the incubation, cells were washed and fresh complete DMEM/F12 medium was added. Selection was initiated 48 h later by adding blasticidin S (5 μg/mL, Invitrogen, Waltham, MA, USA) for 10 days. To produce SH-SY5Y cells expressing green fluorescent protein (GFP), 50% confluent SH-SY5Y cells were infected with LV-GFP (MOI10) and cultured in serum-free DMEM/F12 for 6 h. Cells were washed and fresh complete DMEM/F12 medium was added. Cells with GFP fluorescence were sorted using FACS Aria III (BD Biosciences, San Jose, CA, USA).

### 2.5. Quantitative Polymerase Chain Reaction (qPCR)

Total RNA was extracted from hADSCs using TRIzol Reagent (Invitrogen, Waltham, MA, USA) following the manufacturer’s instructions. Primers and probes specific to 18S ribosomal RNA (18S rRNA), IL2, VEGF, matrix metalloproteinase 2 (MMP2) and TGF-β1 cDNAs were designed using GenScript Online Real-time PCR (TaqMan) Primer Design Tool (GenScript, Piscataway, NJ, USA) and synthesized by Lytech, Moscow, Russia) (Table 1).

hADSC RNA was used as a template for cDNA synthesis using reverse transcriptase (GoScript™ Reverse Transcription System (Promega, Madison, WI, USA)) according to manufacturer’s instructions. TaqMan-based qPCR was performed in MicroAmp 96 well plates (BioRad Laboratories, Hercules, CA, USA) and contained 1 μL cDNA template, 0.3 μL of primers and probe mix (final primer concentration of 300 nM each), 4.7 μL of MilliQ H_2_O and 4 μL of 10x TaqMan-buffer (Lytech, Moscow, Russia), in a final volume of 10 μL. The qPCR was carried out in CFX96 Touch™ Real-Time PCR Detection System (BioRad Laboratories, Hercules, CA, USA) following the protocol: pre-denaturation at 95 °C for 3 min; 45 cycles of denaturation at 95 °C for 10 s, and annealing at 55 °C for 30 s. The gene expression levels were normalized to 18S rRNA levels. Relative quantification was performed by the comparative threshold cycle (ΔΔCT) method.

### 2.6. Immunofluorescence

To analyze IL2 protein expression in native hADSCs and hADSCs-IL2 an immunofluorescence assay was carried out. For this, 1 × 10^3^ native hADSCs and hADSCs-IL2 were seeded on 24-well plate in 400 µL of DMEM/F12. After 24 h, culture medium was removed and cells were fixed with 250 μL of cold methanol for 10 min at RT. Fixed cells were washed 3 times for 5 min each in Tris-buffered saline (TBS; 50 mm Tris, 150 mm NaCl, pH 7.6). Then cells were incubated with primary anti-IL2 antibodies (ab180780, Abcam, 1:100 dilution in TBS) for 1 h at RT. After that cells were washed 3 times for 5 min in TBS and then incubated with secondary antibodies (Goat anti-Rabbit IgG (P & L) Fluorescei (Goat anti-Rabbit IgG (P & L) Fluorescein, #A102FN, American Qualex, San Clemente, CA, USA 1:1000 dilution in TBS) for 1 h at RT. After washing three times for 5 min in TBS, cells were stained with DAPI fluorescent dye (4′, 6-diamidino-2-phenylindole; dilution 1:50,000 in TBS; Invitrogen, Waltham, MA, USA) for 5 min, and washed again. Coverslips were mounted on the slides with a mounting medium (ImmunoHistoMount, Santa Cruz Biotechnology, Santa Cruz, CA, USA). The samples were investigated under a LSM 780 confocal microscope (Carl Zeiss, Jena, Germany) using Zen black 2012 software (Carl Zeiss, Jena, Germany). All samples were imaged in the z-plane using identical confocal settings (laser intensity, gain, and offset).

### 2.7. Western Blot Analysis

Whole cell (1 × 10^6^ cells per extraction) protein lysates were prepared in RIPA buffer containing Halt™ Protease and Phosphatase Inhibitor Cocktail (Thermo Scientific, Waltham, MA, USA). Equal volumes of lysates were electrophoresed on 4–12% sodium dodecyl-sulfate polyacrylamide gel electrophoresis (SDS-PAGE) gradient gels and transferred onto 0.2 μm nitrocellulose membranes (#162-0112, BioRad Laboratories, Hercules, CA, USA) using the trans-blot semi-dry electrophoretic cell transfer system (BioRad Laboratories, Hercules, CA, USA). Membranes were stained overnight at 4 °C with rabbit polyclonal anti-IL2 primary antibody (1:200; #ab180780, Abcam, USA), rabbit polyclonal anti-TGF-β1 (1:200; #sc-130348, Santa Cruz Biotechnology, Santa Cruz, CA, USA), mouse polyclonal anti-VEGF primary antibody (#sc-7269, Santa Cruz Biotechnology, Santa Cruz, CA, USA), mouse polyclonal anti-MMP2 primary antibody (#sc-13594, Santa Cruz Biotechnology, Santa Cruz, CA, USA) or anti-β-actin horseradish peroxidase (HRP)-conjugated antibody (1:1000; #A00730, GenScript, Piscataway, NJ, USA) followed by HRP-conjugated goat anti-rabbit immunoglobulin G (1:2000; #8a0467j, American Qualex Antibodies, San Clemente, CA, USA) or HRP-conjugated goat anti-mouse immunoglobulin G (A3682, Sigma-Aldrich, St. Louis, MO, USA) for 2 h at RT. Proteins were visualized with Clarity™ Western ECL Substrate (#1705061, BioRad Laboratories, Hercules, CA, USA) on a ChemiDoc XRS+ system (BioRad Laboratories, Hercules, CA, USA). The target protein signal intensity was normalised against the housekeeping protein β-actin and relative levels of signal intensity quantified using Image Lab^TM^, version 6.0.1 (Bio-Rad Laboratories, Hercules, CA, USA).

### 2.8. Immunophenotyping

Immune phenotyping of hADSCs was undertaken using antibodies to: CD90 (FITC) (#555595, BD Biosciences, San Jose, CA), CD29 (APC) (#303008, BioLegend, San Diego, CA, USA), CD166 (PE) (#343904, BioLegend), CD44 (PE) (#103024, BioLegend, San Diego, CA, USA), CD73 (APC) (#344006, BioLegend, San Diego, CA, USA), and a negative control (PE) (BD StemflowTM Human MSC Analysis Kit, BD Biosciences, San Jose, CA, USA) including antibodies to CD34, CD11b, CD19, CD45, and HLA-DR. Native hADSCs, hADSCs-BFP and hADSCs-IL2 were trypsinized and washed (2× PBS). Cell aliquots (1 × 10^5^ cells/mL) were incubated with appropriate antibodies for 30 min protected from light at room temperature. Cells were washed once with PBS and analyzed by flow cytometry using FACS Aria III (BD Biosciences, San Jose, CA, USA) and data analyzed using BD FACSDiva™ software version 7.0.

### 2.9. Conditioned Medium Collection

Native hADSCs, hADSCs-BFP and hADSCs-IL2 were seeded at a density of 2 × 10^5^ cells in T75 culture flasks. Conditioned medium (CM) was harvested after 24, 48 and 72 h of cultivation, centrifuged (1500 rpm for 5 min at room temperature) and stored at −80 °C.

### 2.10. Cell Proliferation Assays

To analyze hADSC proliferation, cells were collected at a density of 70–80% confluence, resuspended in DMEM/F12 medium, and then plated at a concentration of 5 × 10^3^ cells in 100 μL of medium per well of a 96-well culture plate and incubated for 24 and 48 h.

To analyze SH-SY5Y, A549 or PC3 proliferation cells were plated at a concentration of 5 × 10^3^ cells in 100 μL of DMEM/F12 medium in a 96-well plate. Cells were incubated for 24 h at 37 °C, 5% CO_2_, the medium was replaced with CM (various dilutions) harvested from the hADSCs at various time points (total volume 100 μL per well) and incubated for 48 h. Proliferation of hADSCs or cancer cells was evaluated using the CellTiter 96^®^ Aqueous Non-Radioactive Cell Proliferation Assay kit (Promega, Madison, WI, USA) following the manufacturer’s instructions. The absorbance in the wells was determined using an Infinite M200Pro (Tecan Trading AG, Mannedorf, Switzerland) reader in a two-wave mode at 490 nm (main) and 630 nm (reference) wavelengths. Two biological replicates were completed for all samples.

### 2.11. Transmission Electron Microscopy

Native hADSCs, hADSCs-BFP and hADSCs-IL2 were fixed in 2.5% glutaraldehyde in PBS for 24 h at 4 °C. Post-fixation, cells were treated with 1% osmium tetroxide for 2 h and subsequently dehydrated using ethanol at 30 to 96% *v/v* concentrations, acetone and then a final treatment in propylene oxide before embedding in Epon 812 resin. After resin polymerization at 37, 45, and 60 °C, samples were cut into ultrathin sections using ultramicrotome (Leica UC7, Leica Biosystems, Wetzlar, Germany). Sections were mounted on copper grids (Sigma-Aldrich, St. Louis, MO, USA, 200 mesh) and contrast agents uranyl acetate and lead citrate were added. Ultrathin sections were examined using a transmission electron microscope (TEM) HT7700 (Hitachi, Tokyo, Japan) at 100 kV.

### 2.12. Cytokine Multiplex Analysis

The Human Chemokine 40-plex Panel (#171ak99mr2, BioRad Laboratories, Hercules, CA, USA) was used to analyze CM samples according to the manufacturer’s recommendations. Human Chemokine 40-plex Panel detects CCL21, CXCL13, CCL27, CXCL5, CCL11, CCL24, CCL26, CX3CL1, CXCL6, GM-CSF, CXCL1, CXCL2, CCL1, IFN-ϒ, IL1β, IL2, IL4, IL6, IL8/CXCL8, IL10, IL16, IP10/CXCL10, I-TAC/CXCL11, MCP-1/CCL2, MCP-2/CCL8, MCP-3/CCL7, MCP-4/CCL13, MDC/CCL22, MIF, MIG/CXCL9, MIP-1α/CCL3, MIP-1δ/CCL15, MIP-3α/CCL20, MIP-3β/CCL19, MPIF-1/CCL23, SCYB16/CXCL16, SDF-1α+β/CXCL12, TARC/CCL17, TECK/CCL25, TNF-α. Fifty microliters of each sample was used to determine cytokine concentration and the collected data was analyzed using a Luminex 200 analyzer with MasterPlex CT control and QT analysis software (MiraiBio division of Hitachi Software San Francisco, CA, USA). Each Bioplex analysis was conducted in triplicate for two biological replicates.

### 2.13. Peripheral Blood Mononuclear Cell (PBMC) Activation

PBMCs were plated in 35 mm non-treated dishes (2 × 10^6^ cells per dish) in 2 mL of CM harvested from hADSCs-IL2, native hADSCs or fresh DMEM/F12. Activation of PBMCs populations was determined at 72 h by flow cytometry using FACS Aria III (BD Biosciences, San Jose, CA, USA), a minimum of 20,000 events were acquired for each sample. The following populations of the immune cells were investigated: late-activated T-cells (CD3^+^, HLA-DR^+^), activated CD8^+^ T-cells (CD8^+^, CD38^+^), early-activated T-cells (CD3^+^, CD25^+^), NK cells (CD3^−^, CD56^+^), NKT-cells (CD3^+^, CD56^+^). PBMCs were washed twice with PBS and resuspended at 1 × 10^5^ cells/mL in PBS before being incubated with antibodies for 30 min in the dark at RT. All samples were stained with allophycocyanin (APC)-conjugated anti-human CD3 (#300312, BioLegend, San Diego, CA, USA) or Alexa Fluor 488-conjugated anti-human CD3 (#2324020, Sony Biotechnology, San Jose, CA, USA), Alexa Fluor 488-conjugated HLA-DR (#IM0463U, Beckman Coulter, USA), Alexa Fluor 647-conjugated anti-human CD56 (NCAM) (#2191564, BioLegend, San Diego, CA, USA), R-phycoerythrin (PE)-Cy7-conjugated anti-human CD25 (#2113060, Sony Biotechnology, San Jose, CA, USA), PE-conjugated anti-human CD8a (#300908, BioLegend, San Diego, CA, USA) and Alexa Fluor 488-conjugated anti-human CD38 (#2117515, Sony Biotechnology, San Jose, CA, USA).

### 2.14. Co-Culture of Cancer Cells and Human Adipose Tissue-Derived Mesenchymal Stem Cells (hADSCs) on Plastic

SH-SY5Y cancer cells were seeded together with native hADSCs, hADSCs-BFP or hADSCs-IL2 at a 1:1 ratio (using 5 × 10^3^ cells of each cell type per well) in 96-well plate, 6 replicates of each combination were used [40]. Cells were co-cultured in DMEM/F12 medium for 72 h. Cell proliferation was determined using CellTiter 96^®^ Aqueous Non-Radioactive Cell Proliferation Assay kit (Promega, Madison, WI, USA) as previously described [41].

Apoptosis and necrosis of SH-SY5Y-GFP after co-culture with native hADSCs, hADSCs-BFP or hADSCs-IL2 were analyzed using 5 × 10^4^ cells arrayed in triplicate in 6-well plates. After 72 h of co-culture, SH-SY5Y-GFP cells were separated from native hADSCs, hADSCs-BFP or hADSCs-IL2 using a FACS Aria III (BD Biosciences, San Jose, CA, USA). After separation, 5 × 10^4^ sorted SH-SY5Y cells were immediately stained to detect apoptosis and necrosis using an APC Annexin V Apoptosis Detection Kit with PI (#3804660, Sony Biotechnology, San Jose, CA, USA) according to the manufacturer’s protocol. Stained cells were analyzed by flow cytometry using a FACS Aria III (BD Biosciences, San Jose, CA, USA), a minimum of 15,000 events were acquired for each sample.

### 2.15. Co-Culture of Cancer Cells and hADSCs on Matrigel

Matrigel was thawed on ice, 200 µL of Matrigel (#356235, BD Biosciences, San Jose, CA, USA) was placed into each compartment of a 12-well plate on ice, using cooled pipette tips. For Matrigel polymerization, plates were incubated at 37 °C for 30 min before seeding of the cells.

SH-SY5Y cells and hADSCs were labelled with DiO (green) and DiD (red) dyes, respectively, using Vybrant Multicolor Cell-Labeling Kit (#V-22889, Invitrogen, Waltham, MA, USA) for general cell membrane labelling, according to the manufacturer’s instructions. 5 × 10^5^ hADSCs and 2 × 10^6^ SH-SY5Y were resuspended in 500 μL of serum-free medium and stained with 2.5 μL of the corresponding dye. hADSCs were incubated with dye for 15 min, SH-SY5Y were incubated with dye for 5 min at 37 °C, after which the cell suspension was centrifuged at 1500 rpm for 5 min and washed three times with full DMEM/F12.

Stained cells were seeded on a Matrigel-coated 12-well plate in a 1:1 ratio (using 5 × 10^3^ cells of each cell type per well) in 1 mL of DMEM/F12. Self-organization of cancer cells and hADSCs in co-culture was assessed within 120 h using phase-contrast and fluorescence microscopy with an Axio Observer.Z1 inverted microscope (CarlZeiss, Jena, Germany) and Axio Vision software version 4.8.

### 2.16. Statistical Analysis

Statistical analysis was achieved using GraphPad Prism 7 software (GraphPad Software, San Diego, CA, USA), one-way analysis of variance (ANOVA) followed by Tukey honest significant difference (HSD) post-hoc comparisons test. Significant probability values are denoted as * *p* < 0.05, ** *p* < 0.01 and *** *p* < 0.001, **** *p* < 0.0001.

## 3. Results

### 3.1. hADSCs-IL2 Overexpress Interleukin 2 (IL2) mRNA and Protein and Retain a Mesenchymal Stem Cell Phenotype

MSCs were isolated from adipose tissue and genetically modified with LV-IL2 (hADSCs-IL2) or LV-BFP (hADSCs-BFP). Expression of *IL2* gene mRNA was increased by 1000 times in hADSCs-IL2 compared to native hADSCs and hADSCs-BFP (Figure 1a). Overexpression of IL2 protein within hADSCs-IL2 was confirmed by immunofluorescence assay and western blot analysis (Figure 1b,c). Genetic modification failed to have a significant effect on the proliferative activity of hADSCs-IL2 and hADSCs-BFP (Figure 1c). Immunofluorescence analysis of native hADSCs, hADSCs-BFP and hADSCs-IL2 determined that the majority of native and genetically modified hADSCs shared the same pattern of surface antigen expression, which was similar to that commonly detected on human MSCs. The percentage of cells expressing these antigens in native hADSCs was as follows: CD29 98.8% ± 0.2%, CD44 99.6% ± 0.4% cells, CD73 55.3% ± 3.1%, CD90 55.4% ± 6.4%, and CD166 92.4% ± 4.8%. Native hADSCs had a lower expression of the hematopoietic cell surface markers CD34, CD11b, CD19, CD45, and HLA-DR (3.4% ± 1.1% positivity) (Figure 2); whilst hADSCs-IL2 expressed CD29, CD44, CD73, CD90, CD166, in 98.1% ± 0.9%, 99.2% ± 0.8%, 46.7% ± 4.8%, 53.1% ± 3.4% and 94.7% ± 3.6% of cells respectively. Similar to native hADSCs, hADSCs-IL2 were negative for hematopoietic cell surface markers CD34, CD11b, CD19, CD45, and HLA-DR (0.8% ± 0.2% positivity) (Figure 2). hADSCs-BFP expressed CD29, CD44, CD73, CD90, CD166, in 99.1% ± 0.9%, 99.3% ± 0.5%, 48.2% ± 3.7%, 55.1% ± 4.7% and 91.3% ± 5.1% of cells respectively. hADSCs-BFP were also negative for hematopoietic cell surface markers CD34, CD11b, CD19, CD45, and HLA-DR (1.3% ± 0.8% positivity) (Figure 2a).

Like native hADSCs, the genetically modified hADSCs-IL2 maintained the ability to differentiate into adipogenic, osteogenic and chondrogenic lineages (Figure 2b). Together, these data indicate that hADSCs-IL2 retain the phenotype of a mesenchymal stem cell, including the differentiation properties of this cell population.

### 3.2. IL2 Expression Affects the hADSC Ultrastructure

Ultrastructure analysis of native hADSCs, hADSCs-BFP and hADSCs-IL2 was achieved using TEM. In native hADSCs, the rough endoplasmic reticulum (ER) formed elongated cisterns with defined ribosomes and medium electron dense internal contents (Figure 3a). Changes in the rough ER structure were detected in all genetically modified cells (hADSCs-IL2 and hADSCs-BFP), with cells showing a strong expansion of ER cisterns and irregular shape (round or oval). In addition, the rough ER cisterns had a high electron dense content (Figure 3b,c). In some modified cells, ribosomes were absent on the surface of the membrane cisterns of the ER and/or possessed rounded endosomes with a high electron density in the cytoplasm (Figure 3d). In general, the cytoplasm of hADSCs-IL2 and hADSCs-BFP had a high number of organelles, such as endosomes, ER, and the active Golgi complex (GC) and a large number of outgoing vesicles and multivesicular bodies. The nuclei of these modified cells also contained euchromatin and heterochromatin clusters near the wall, the karyolemma was often blurred and nuclear pores were visible (Figure 3e).

### 3.3. IL2 Overexpression Leads to a Modest Change in Cytokine Production

The levels of cytokines CX3CL1, CXCL6, IL8, CCL13, CCL15 and CCL20 were determined in the CM of native hADSCs, hADSCs-BFP or hADSCs-IL2. CM levels of these cytokines were reduced (1.5–2 times) in hADSCs-IL2 cultured over 72 h compared to native hADSCs and hADSCs-BFP (Figure 4c). In addition to the above cytokines, changes in IL6 and IL8 secretion between the hADSC groups were also determined. After 24 h of incubation, the level of secreted IL6 detected in the CM was approximately the same across all hADSC groups (native hADSCs, hADSCs-BFP or hADSCs-IL2) (Figure 4a). However, at 48 h and 72 h, IL6 levels in hADSCs-IL2 CM were significantly lower (4535.2 ± 576.1 pg/mL and 27,925.7 ± 5796.7 pg/mL at 48 and 72 h respectively) than for native hADSCs (12,142.9 ± 1675.6 pg/mL and 27,925.7 ± 5796.7 pg/mL) and hADSCs-BFP (9242.2 ± 21.7 pg/mL and 27,925.7 ± 5796.7 pg/mL) (Figure 4b,c). Like IL6, IL8 was also detected at relatively high levels in media conditioned for 24 h, 11,666.5 ± 496.4 pg/mL in native hADSCs, 9857.5 ± 3.8 pg/mL in hADSCs-BFP and 8209.6 ± 3563.2 pg/mL in hADSCs-IL2 (Figure 4a). However, in media conditioned for both 48, the concentration of secreted IL8 was decreased in hADSCs-BFP (3838.4 ± 7.6 pg/mL) and in hADSCs-IL2 (1665.3 ± 453.2 pg/mL) compared to native hADSCs (11,314.3 ± 250.0 pg/mL) (Figure 4b). Whilst the level of secreted IL8 decreased across all hADSC groups at 48 and 72 h when compared to the levels detected at 24 h. The decreases seen in hADSCs-IL2 (1075.0 ± 10.0 pg/mL) was two-fold lower compared to native hADSCs (2158.5 ± 104.6 pg/mL) and hADSCs-BFP (2038.1 ± 10.7 pg/mL) at 72 h (Figure 4c).

Finally, secretion of tumor necrosis factor α (TNF-α) was increased by 1.5-fold in hADSCs-IL2 (6.8 ± 0.2 pg/mL) compared to native hADSCs (4.7 ± 0.2 pg/mL) and hADSCs-BFP (4.6 ± 0.1 pg/mL) after 72 h of cultivation (Figure 4c).

### 3.4. hADSC-IL2 Conditioned Medium (CM) Can Stimulate the Proliferation of Tumor Cells

The effect of the CM harvested from native hADSCs, hADSCs-BFP or hADSCs-IL2 on SH-SY5Y (neuroblastoma), PC3 (prostate cancer) and A549 (lung adenocarcinoma) tumor cell proliferation in vitro was investigated. CM was harvested after 24, 48 and 72 h of culture from native hADSCs, hADSCs-BFP or hADSCs-IL2. Cancer cells were plated at a concentration of 5 × 10^3^ cells/100 μL of medium per well in a 96-well culture plate. After 24 h, the medium was replaced with CM from native hADSCs, hADSCs-BFP or hADSCs-IL2 harvested at the 24, 48 or 72 h time points. CM was either applied undiluted (100% CM) or at a 50:50 dilution of CM with DMEM/F12. Cancer cells were grown in CM for 48 h before analysis of their proliferation activity. Growth of SH-SY5Y cells in undiluted (100%) CM obtained from hADSCs-IL2 resulted in the increase in SH-SY5Y proliferation by 65% when using 24-h CM (*n* = 6, *p* < 0.01), by 10% with 48-h CM (*n* = 6, *p* < 0.05) and 30% with 72-h CM (*n* = 6, *p* < 0.05) when compared to SH-SY5Y cells grown in CM obtained from hADSCs and hADSCs-BFP across the same time points (Figure 5). The effect of hADSCs-IL2 CM on PC3 cell proliferation was more variable, showing decreased proliferation of PC3 cells by 13% when exposed to media conditioned by hADSCs-IL2 for 24 h (*n* = 6, *p* < 0.01), but an increase in proliferation by 16% (*n* = 6, *p* < 0.01) when treated with media conditioned for 72-h, when compared to hADSCs and hADSCs-BFP (Figure 5). CM had no significant effect on A549 proliferative activity regardless of the cells of origin. Interestingly, the proliferation of PC3 and A549 cells was greater when grown in fresh DMEM/F12 alone than when exposed to CM from any hADSC group and/or time point. Only SH-SY5Y cells demonstrated greater proliferation in CM, when media was conditioned by hADSCs-IL2 for 72 h (Figure 5).

### 3.5. hADSCs-IL2 Can Promote Angiogenesis and Increase Tumor Cell Invasion

It is known that MSCs are key participants of the tumor microenvironment and, in addition to being able to suppress various tumor growth, they can promote tumor growth and metastasis, as well as stimulate neovascularization by expressing multiple pro-angiogenic and trophic factors such as VEGF, IL8, TGF-β, EGF, and PDGF [17]. In this context, increased pro-oncogene expression in response to further modifications of MSCs could cause a significant potential problem when using modified MSCs for patient treatment. Therefore, we sought to determine the effect of IL2 overexpression on the abundance of *VEGF*, *MMP2* and *TGF-β1* genes, known to have pro-oncogenic and pro-angiogenic properties [42,43,44] in hADSCs-IL2.

Using quantitative PCR of transcript levels, it was shown that, *VEGF* gene expression in hADSCs-IL2 was increased 2.3-fold compared to native hADSCs (*n* = 3, *p* < 0.0001) and 1.5-fold compared to hADSCs-BFP (*n* = 3, *p* < 0.0001) (Figure 6a). Similarly, *MMP2* expression was increased 2.3-fold in the hADSCs-IL2 compared to native hADSCs (*n* = 3, *p* < 0.0001) and 1.5-fold compared to hADSCs-BFP (*n* = 3, *p* < 0.0001) (Figure 6b). Finally, *TGF-β1* gene expression was also confirmed to increase in hADSCs-IL2, 1.6 and 1.2-fold compared to native hADSCs and hADSCs-BFP (*n* = 3, *p* < 0.0001), respectively (Figure 6c). However, Western blot analysis of VEGF protein expression showed that there was no increase in VEGF protein expression in hADSCs-IL2 compared to hADSCs-BFP or native hADSCs (relative expression of 93%, 96% and 100%, respectively). The relative expression of MMP2 protein was slightly increased (112%) in hADSCs-IL2 compared to hADSCs-BFP (108%) and native hADSCs (100%). A slight increase could also be observed in TGF-β1 protein expression in hADSCs-IL2 (114%) compared to hADSCs-BFP (106%) and native hADSCs (100%) (Figure 6a–c). These data suggest that the over-expressed IL2 impacts the transcription and protein expression of factors that modify the tumor microenvironment, such as MMP2 and TGF-β1. However, as in the case of *VEGF* changes in transcription are not always seen at the protein level.

### 3.6. hADSCs-IL2 Activates Mononuclear Blood Cells In Vitro

To determine if activation of PBMCs changes the expression of the leukocyte surface markers, an analysis of a number of activation markers on human PBMCs was used to determine the immunogenic effects of hADSCs-IL2 on T and NK cells.

De novo expression of HLA-DR is characteristic for the late-activation stages (3–5 days) of T-cells [45]. We found that 6.3 ± 0.4% of PBMCs cultured in the presence of hADSCs-IL2 CM express CD3/HLA-DR which is typical for late T-cell activation. The percentage of CD3^+^/HLA-DR^+^ PBMCs cultured in CM from native hADSCs and DMEM/F12 medium was significantly lower, 2.4 ± 0.3% and 2.8 ± 0.6%, respectively (*n* = 3, *p* < 0.01) (Figure 7a).

CD25, α subunit of IL2 receptor, is a marker of an early (12 h) T-cell activation [46]. The number of CD3^+^/CD25^+^ cells was 7.1 ± 0.3% in PBMCs cultured in hADSCs-IL2 CM, similar to the numbers found when PBMCs were cultured in CM from native hADSCs or DMEM/F12 medium alone (6.9 ± 0.4% and 8.0 ± 0.5%, respectively) (Figure 7b).

Leukocytes expressing T-cell marker CD3 and the NK cell marker CD56, are identified as NKT-cells. NKT-cells are cytotoxic against tumor cells and independent of T-cell receptor activation [47]. We have shown that 3.0 ± 0.3% of PBMCs grown in CM from hADSCs-IL2 were positive for CD3/CD56 markers, compared to just 1.3 ± 0.5% and 1.6 ± 0.4% of PBMCs incubated in CM from native hADSCs and DMEM/F12 medium respectively (*n* = 3, *p* < 0.01) (Figure 7c).

CD38 expression is associated with late (1 day) activation and proliferation of cytotoxic CD8+ T lymphocytes (CTLs) [45,48]. Analysis of the CD8^+^ and CD38^+^ expression revealed that 33.4 ± 1.4% of PBMCs expressed these receptors when cultured in hADSCs-IL2 CM. Significantly lower numbers of CD8^+^/CD38^+^ cells were found when PBMCs were cultured in CM from native hADSCs and DMEM/F12, 13.7 ± 1.8% and 12.4 ± 1.2%, respectively (*n* = 3, *p* < 0.01) (Figure 7d). These data confirm our previous observation that hADSCs-IL2 CM affects the lymphocyte activation pattern, increasing the population of late-activated T cells.

Expression of the neural cell adhesion molecule (NCAM), also known as CD56, in the absence of CD3 is characteristic for NK cells. NK cells can be divided into two populations: CD56^dim^ and CD56^bright^ NK cells [49]. We found that the number of CD56^+^/CD3^−^ NK cells was higher in PBMCs cultured in hADSCs-IL2 CM (8.8 ± 0.5%) compared to native hADSCs CM and DMEM/F12 (4.3 ± 0.7% and 3.1 ± 0.3% respectively) (*n* = 3, *p* < 0.01) (Figure 7e). The majority of the NK cells cultured in hADSCs-IL2 CM were CD56^bright^ (72.5 ± 1.3% of the entire NK cell population), while the higher proportion of CD56^dim^ NK cells was found in PBMCs cultured in CM from native hADSCs and DMEM/F12 (78.5 ± 0.7% and 70.0 ± 0.9% of the entire NK cell population, respectively) (Figure 7e).

### 3.7. Co-Culture of hADSCs-IL2 Reduces Proliferation and Mediates Apoptosis of SH-SY5Y

Our data demonstrate that CM harvested from hADSCs-IL2 increases SH-SY5Y cell proliferation. However, cells can communicate by paracrine mechanisms or through direct cell–cell contact when co-cultured. Therefore, we sought also to determine the effect of the hADSC co-culture on survival and proliferation of SH-SY5Y tumor cells. Neuroblastoma SH-SY5Y cells or SH-SY5Y-GFP cells were co-cultured with native hADSCs, hADSCs-BFP or hADSCs-IL2 at a 1:1 ratio. Analysis of proliferative activity demonstrated that co-culture of SH-SY5Y with hADSCs-IL2 caused a significant decrease (~20%) in cell proliferation (cumulative proliferation index of all cell populations in co-culture, 78.1 ± 7.9%) compared to the co-culture of tumor cells with native hADSCs (100.0 ± 3.3%) or hADSCs-BFP (107.7 ± 6.6%) (*n* = 6, *p* < 0.0001) (Figure 8a).

In addition to proliferation, tumor cell viability was determined by detection of apoptotic and necrotic tumor cells. SH-SY5Y-GFP cells were separated from hADSCs using FACS (Aria III, BD Biosciences, San Jose, CA, USA). Analysis of the number of apoptotic and necrotic SH-SY5Y cells after co-culture with hADSCs-IL2 revealed a modest but significant decrease in cell viability (percentage of non-apoptotic cells 86.3 ± 0.3%) compared to cells co-cultured with native hADSCs (89.6 ± 1.1%) (*n* = 3, *p* < 0.05), however there was no significant difference between hADSCs-IL2 and hADSCs-BFP (86.2 ± 1.9%) (Figure 8b). The same results were observed for the number of apoptotic and necrotic SH-SY5Y cells. The percentage of apoptotic SH-SY5Y cells, after co-culture with hADSCs-IL2, was significantly increased (3.0 ± 0.4%), when compared to co-culture with native hADSCs (2.3 ± 0.1%) (*n* = 3, *p* < 0.05), but there was no significant difference between SH-SY5Y cells co-cultured with hADSCs-IL2 and hADSCs-BFP (3.3 ± 0.1%) (Figure 8c). The percentage of necrotic SH-SY5Y cells, when co-cultured with hADSCs-IL2, was significantly increased (3.2 ± 0.2%), in comparison with co-culture with native hADSCs (2.3 ± 0.3%) (*n* = 3, *p* < 0.01), but there was no significant difference between SH-SY5Y cells co-cultured with hADSCs-IL2 and hADSCs-BFP cells (2.7 ± 0.2%) (Figure 8d).

In order to evaluate the organization and interaction of stem and tumor cells, native hADSCs, hADSCs-BFP or hADSCs-IL2 and SH-SY5Y cells were co-cultured on the Matrigel matrix for 120 h. After vital staining of hADSCs with Vybrant DiD dye and SH-SY5Y cells with Vybrant DiO dye, all the cells had red or green fluorescence, respectively, which made it possible to distinguish these cell types in co-culture. An interesting self-organization of hADSCs and SH-SY5Y in co-culture was observed (Figure 9). After 48 h, the cells organized into individual cellular aggregates with flying saucer-like architecture, in which the core consisted of hADSCs that were surrounded by flat aureole of SH-SY5Y cells. Throughout the observation period, SH-SY5Y cells actively proliferated around stem cells, the proliferation rate of tumor cells was significantly higher than the proliferation rate of hADSCs, which led to the compression of hADSC cultures. It is interesting to note that in a co-culture with hADSCs-IL2 the inhibition of SH-SY5Y tumor cell growth was clearly observed compared to co-cultures of SH-SY5Y cells with native hADSCs or hADSCs-BFP (Figure 9).

## 4. Discussion

The use of MSCs as a vehicle for the delivery of therapeutic agents can increase the effectiveness of IL2-based immunotherapy by stimulating an antitumor immune response or directly inhibiting tumor growth. Our results showed that hADSCs-IL2 overexpressed both *IL2* gene mRNA and IL2 protein. At the same time, genetic modification did fail to have a significant effect on the proliferative activity and immunophenotype of hADSCs-IL2 and hADSCs-BFP compared to native hADSCs. Like native hADSCs, the genetically modified hADSCs-IL2 were able to differentiate into adipogenic, osteogenic and chondrogenic lineages. However, overexpression of IL2 did cause significant changes in the hADSCs-IL2 ultrastructure, where an increased size and density of rough ER, and the Golgi apparatus were noted. However, significant changes in the ultrastructure of GC and ER in hADSCs-BFP were also seen when compared to native hADSCs. Therefore, these changes may be linked to recombinant protein overproduction rather than IL2 directly. Data produced by Hussain et al., demonstrate that ER stress can occur in response to increased amounts of unfolded and misfolded proteins, a common phenomenon associated with protein over-expression [50]. To cope with this stress, the ER induces several mechanisms, including the unfolded protein response (UPR) [51], which in turn can inhibit protein synthesis, up-regulate the capacity of the cell to fold proteins and also increase the production of lipids [52]. These newly produced lipids insert into the ER membrane which results in ER expansion and the decrease in ER stress [53].

Changes in cytokine profiles were also determined as a result of IL2 over-expression. The release of CX3CL1, CXCL6, IL8, CCL13, CCL15 and CCL20 was significantly decreased (by 1.5 times) in hADSCs-IL2. These cytokines have been shown to modulate the anti-tumor immune response, the outcome of which can vary from protection against the cancer to promoting the growth of the malignancy. For example, CX3CL1 was shown to enhance the antitumor response by activating NK cells and T-cells [54]. Whilst CCL13 triggers monocytes, T-cells and immature dendritic cells (DCs) to migrate towards the tumor [55]. However, several cytokines have been shown to promote tumor growth through increasing angiogenesis, including CXCL6 and IL8 [56,57]. Increased CCL15 and CCL20 expression has been reported in a number of cancers and is associated with a poor prognosis [58,59,60].

IL6 secretion was high in all hADSCs (20,000–27,000 pg/mL after 72 h), consistent with previous reports of human adipose tissue-derived MSCs [61]. It has been shown that high levels of IL6 secretion can lead to increased tumor invasion [62], proliferation and resistance to chemotherapy [63]. In particular, high secretion of IL6 in MSCs can activate the IL6/STAT3 signaling pathway and promote cell invasion in hepatocellular carcinoma cells [62]. Although IL6 secretion after 24 h in all hADSCs was approximately at the same level, cytokine release was significantly lower at 48 and 72 h in hADSCs-IL2 compared to native hADSCs and hADSCs-BFP. This reduction in IL6 secretion, may allow hADSCs-IL2 to have an inhibitory effect on tumor growth and proliferation compared to native hADSCs. Like IL6, IL8 secretion was also high in both native and genetically modified hADSCs (9000–11,000 pg/mL) after 24 h of cultivation. In addition to the fact that MSCs-secreted IL8 can stimulate the proliferation and angiogenesis of a number of tumors [57,64], this protein is also an important participant of the epithelial-mesenchymal transition (EMT) of cancer cells [65]. It has been shown that IL8 secreted by tumor cells undergoing EMT can enhance tumor progression by inducing adjacent epithelial tumor cells to also undergo EMT [66]. However, it is worth noting that the level of IL8 was significantly decreased in all hADSCs after 72 h of cultivation (a 5-fold reduction compared to IL8 levels detected at 24 h). In addition, IL8 secretion in hADSCs-IL2 was 2 times lower compared to native hADSCs and hADSCs-BFP after 72 h of cultivation, which is likely due to overexpression of IL2, but the molecular mechanisms of this effect require further investigation. TNF-α secretion was also found to be higher (1.5-fold) in hADSCs-IL2 compared to native hADSCs and hADSCs-BFP. TNF-α has been shown to inhibit tumor progression in mice [67]. However, its effect on tumor angiogenesis appear to depend on cytokine concentration. TNF-α is produced in small quantities (pg) by tumor cells and the tumor microenvironment, and this low level expression is usually associated with poor prognosis [68,69]. Furthermore, TNF-α concentrations in the range of 0.5–50 ng/mL, can induce chemotaxis of endothelial cells of adrenal capillaries in vitro and cause formation of a branching capillary-like tubular structures. However, blood vessel formation is abrogated by higher concentrations of TNF-α (500 ng/mL) [70]. Therefore, effects of hADSCs-IL2 on tumor growth would depend on the cumulative concentration of TNF-α secreted. The levels of TNF-α secreted by hADSCs-IL2 could be considered to be within the low range and likely to promote the tumorigenic microenvironment. However, these effects would need to be determined in vivo and per dose of hADSCs-IL2 used.

In addition to changes in cytokine expression, overexpression of *IL2* also led to changes in the expression of a range of other genes, including oncogenes. We found that hADSCs-IL2 also increased transcription of *VEGF, MMP2* and *TGF-β1*. In confirmation of our findings, previous published studies have found *VEGF* expression to be increased by more than 60% in Bcl-2 transfection-modified hypoxic MSCs and VEGF secretion increased in genetically modified MSCs when cultured in hypoxia [71,72]. We also analyzed the expression level of VEGF, MMP2 and TGF-*β1* at the protein level in hADSCs. Increased expression of VEGF protein in MSCs can be a potential problem, since the secretion level of this protein in MSCs is high, and it was noted that MSCs can stimulate angiogenesis of a number of tumors, including pancreatic carcinoma and hepatocellular carcinoma [73,74]. However, the increase in *VEGF* gene expression was detected at the mRNA level only, and similar changes in protein expression were not detected. It remains unclear whether the increases in *VEGF* gene expression subsequently translate into changes in VEGF protein levels or whether other mechanisms, such as translation inhibition and RNA degradation, act to suppress IL2-induced oncogene expression at the protein level.

Whilst VEGF did not show a comparable increase at the protein level, both MMP2 and TGF-β1 showed increased protein translation as well as mRNA transcription. MMP2 and TGF-β1 both play an important role in EMT of tumor cells [75]. Tumor progression, in particular, tumor metastasis can also be supported by MMP2, which facilitates tissue penetration by cancer cells [43]. MMP-2 also plays an important role in regulation of MSCs migration and proliferation [76]. Therefore, the increased *MMP2* expression we detected in hADSCs-IL2 compared to native hADSCs and hADSCs-BFP, could promote MMP2-mediated migration [77], proliferation [78] and angiogenesis [79]. Interestingly, *TGF-β1* expression was also higher in hADSCs-IL2 as compared to native hADSCs and hADSCs-BFP. TGF-β plays a dual role in tumor progression acting as both a tumor suppressor and pro-oncogenic factor [44]. TGF-β also has the ability to stimulate EMT as well as support the mesenchymal phenotype of tumor cells that have passed through EMT [80,81]. However, as we found increased expression of *TGF-β1* in hADSCs-BFP as well as hADSCs-IL2, increased *TGF-β1* may result from any genetic modification of hADSCs rather than being IL2 specific. The pro or anti-tumor effects of increased *TGF-β1* in hADSCs, regardless of the trigger, remains unclear.

Overall, the expression level of *VEGF*, *MMP2* and *TGF-β1* was significantly higher in hADSCs-IL2 compared to hADSCs-BFP, suggesting that IL2 expression has an effect on the transcription activity of these cells. While it is established that increased *VEGF* and *MMP2* expression can stimulate blood vessel growth and metastases and, thereby, contribute to tumor progression, TGF-β1 can have both, tumor-suppressing or pro-oncogenic effects [42,43,44]. In addition, both MMP2 and TGF-β1 proteins as well as IL6 and IL8, whose secretion levels were high in hADSCs-IL2, are important participants in EMT of tumor cells. Based on this data, we hypothesize that activation of these tumor promoting genes and increased expression of MMP2 and TGF-β1 proteins in hADSCs-IL2 could have adverse effects on tumor growth when used for the treatment of cancer patients and requires additional investigation.

This hypothesis was somewhat tested by examining the effect of molecules secreted by hADSCs-IL2 into CM on tumor cell growth, which were seen to have variable effects on the viability of the cancer cells. Whilst A549 tumor cells were unaffected by hADSCs-IL2 CM, the proliferation of SH-SY5Y was significantly increased, whilst the proliferation of PC3 cells was modified both positively and negatively dependent upon the length of time hADSCs-IL2 had conditioned the media. The ability of IL2 to inhibit the proliferation of PC3 cells has been reported in the investigation where the combination of IL2 with trichosanthin led to a stronger inhibition of cell growth compared to trichosanthin alone [82]. Interestingly, cancer cell proliferation in fresh DMEM/F12 was higher compared to cells cultured in CM with the exception of SH-SY5Y cells grown in hADSC CM harvested after 72 h of conditioning. This reduced proliferation of PC3 and A549 cells in CM compared to DMEM/F12 could be explained by depletion of the nutrients within media during hADSCs culture, although we cannot rule out the build-up of byproducts that inhibit cell growth. In turn, the stimulation of SH-SY5Y proliferation after cultivation in hADSC CM is consistent with previous investigations which describe how the hADSC secretome can stimulate proliferation and promote survival of SH-SY5Y cells by secreting neuroprotective factors such as brain-derived neurotrophic factor (BDNF), glial-derived neurotrophic factor (GDNF), beta-nerve growth factor (β-NGF) and insulin-like growth factor-1 (IGF-1) [83,84]. However, when comparing the effect of CM and physical co-culture of the tumor cells with modified hADSCs, the proliferation of SH-SY5Y cells was significantly decreased (over 20%) after co-culture with hADSCs-IL2 on plastic compared with control co-cultures. A slight increase in the number of apoptotic and necrotic SH-SY5Y cells was also found when co-cultured on plastic with hADSCs-IL2 as compared to native hADSCs, but not hADSCs-BFP, which suggests that hADSCs-IL2 itself did not induce the death of neuroblastoma cells and their antitumor effect was due to the ability to suppress SH-SY5Y cell proliferation. In order to evaluate the interaction of hADSCs-IL2 with neuroblastoma cells and confirm the ability of hADSCs-IL2 to suppress SH-SY5Y proliferation in co-culture, we decided to culture tumor cells and hADSCs on a thin layer of Matrigel matrix (BD Biosciences, San Jose, CA, USA), which is rich in extracellular matrix (ECM) components and allows three-dimensional cell culture to be simulated [85]. Matrigel is a useful tool for studying tumor cell growth and their interaction with other types of cells of TME, as far as cancer cell cultures on Matrigel are characterized by multicellular three-dimensional growth and altered cell morphology, proliferation and differentiation [86,87]. It has been previously reported that hMSCs form capillary-like structures on Matrigel [88], while SH-SY5Y cells represent the morphology they have when cultured on plastic [89]. We used Matrigel to evaluate the interaction of SH-SY5Y cells and hADSCs in co-culture. After 48 h, the cells organized into individual cellular aggregates with flying saucer-like architecture, in which the core consisted of hADSCs that were surrounded by flat aureoles of SH-SY5Y cells. Tumor cells were actively proliferating around the stem cell core, the size of individual aggregates was rapidly increasing, which ultimately led to their fusion. At the same time, hADSCs were proliferating slowly, and the core size remained unchanged. The same organization of SH-SY5Y and human bone marrow-derived MSCs has been shown previously [90]. It was interesting that the proliferation rate of tumor cells in co-culture with hADSCs-IL2 was significantly lower in the first 72 h, compared with native hADSCs and hADSCs-BFP. However, when the individual aggregates reached large sizes, the SH-SY5Y cell proliferation rate had increased. We think that this is due to the fact that the proliferation of tumor cells distant from the nucleus was no longer controlled by stem cells, which suppressed their proliferation by direct contact.

Previously reported data also shows that IL2 can directly inhibit cancer cell growth in vivo and in vitro. For example, IL2 was shown to directly inhibit the growth of melanoma cells [91]. Local therapy with IL2 and human lymphokine-activated killer (LAK) cells also significantly reduced the growth of tumors in a nude mouse model of human squamous-cell carcinoma of the head and neck [92]. The efficacy of IL2 in neuroblastoma therapy has also been investigated. Intratumoral injection of human fibroblasts genetically modified to express IL2 showed a significant therapeutic effect with reduced growth or complete eradication of tumors in mice with Neuro-2A neuroblastoma, which was associated with extensive leukocyte infiltration [93]. In clinical trials the use of IL2 can result in controversial effects in the combined immunotherapy [94,95]. For example, addition of subcutaneous IL-2 to immunotherapy with chimeric anti-GD2 monoclonal antibody dinutuximab beta did not improved outcomes in patients with high-risk neuroblastoma [94]. Another clinical trial has demonstrated significantly improved outcomes in patients with high-risk neuroblastoma after the combined treatment with ch14.18 antibody to isotretinoin and IL2 or granulocyte macrophage-colony stimulating factor (GM-CSF) [95]. The diverse effects of CM harvested from hADSCs-IL2 versus direct co-culture with hADSCs-IL2 on proliferation of SH-SY5Y could be explained by the level of biologically active molecules in CM generated by a large number of hADSCs-IL2 cells (compared to the number of hADSCs-IL2 cells in co-culture) or by direct cell-cell inhibitory effects of hADSCs-IL2 due to changes in surface protein expression or other cellular factors altered by IL2 overexpression on SH-SY5Y in co-culture. It remains to be fully elucidated why hADSCs-IL2 CM increases the proliferation of SH-SY5Y tumor cells, yet the physical presence of hADSCs-IL2 in co-culture has a negative effect on tumor cell viability. Taken together, the data from assays using CM and co-culture, support both paracrine pro-actions and potentially negative direct cell-cell mechanisms occurring.

IL2 plays a critical role in immune system activation that could provide a useful way to stimulate anti-tumor immune response [96]. To determine hADSCs-IL2 immunogenic effects on leukocyte surface marker expression, human PBMCs were analyzed for a number of activated T cell and NK cell markers.

De novo expression of the HLA-DR marker is characteristic of the late-activated (3–5 days) T-cells [45]. It has been established that endogenous IL2 can increase the expression of HLA-DR on CD3^+^ T-cells in vitro [46]. Increased expression of the late-activated T-cell marker HLA-DR in PBMCs cultured in the CM from hADSCs-IL2 as compared to native hADSC CM was demonstrated. This suggests that hADSCs-IL2 can affect lymphocyte activation, increasing the population of the late activated T cells. High HLA-DR expression on T-cells from breast cancer patients correlates with a good response to neoadjuvant chemotherapy treatment (NACT) [97]. Therefore, hADSCs-IL2 could be considered a useful addition in combination with chemotherapy.

Significantly large numbers of leukocytes expressing CD38, the CTL activation marker, were found in PBMCs cultured in hADSCs-IL2 CM compared to that from native hADSCs. CD38 expression is associated with late (1 day) activation and proliferation of cytotoxic CD8^+^ T lymphocytes (CTLs) [45,48]. Activation of CTLs is essential for antitumor defense as they are the key leukocytes mediating cancer cell death [98].

CD25 is a marker of an early (12 h) activation of T-cells [46]. It was shown that the recombinant IL2 treatment could increase CD25 expression on T-cells in a dose dependent manner in vitro [99]. However, expression of the early-activated T cell marker CD25 was lower in PBMCs cultured in the hADSCs-IL2 CM as compared to native hADSC CM. However, we may have been limited in our ability to detect changes in expression of this early T-cell activation marker, since we were measuring CD25 at a late time point (after 72 h in culture).

NK cell populations which express NCAM (CD56) in the absence of CD3 can be divided into two populations: CD56^dim^ and CD56^bright^ NK cells [49]. CD56^dim^ NK cells are cytotoxic and commonly found in peripheral blood, while CD56^bright^ cells are located in the lymph nodes and play a regulatory role, secreting IFN-γ [100]. NK cells can be activated by IL2 and gain tumor-cell killing capacity [101]. Furthermore, Dubois et al. have shown that IL15 can increase the number of CD56^dim^ and CD56^bright^ NK cells as well as potentiate the cytotoxic capacity of CD56^bright^ cells [102]. NK cell number was increased in PBMCs cultured in hADSCs-IL2 CM, with the majority of cells expressing CD56^bright^ receptor. A high level of tumor infiltrating NK cells is often associated with increased cancer patient survival in periampullary adenocarcinoma, colorectal carcinoma and gastric carcinoma [103,104,105]. The CD56^bright^ NK cell population has been shown to be predominant among patients within breast, melanoma, colon cancer and non-small cell lung cancer, having pro-angiogenic effects and, thereby, promoting tumorigenesis [106,107]. However, stimulation with endogenous ILs can increase the number of CD56^bright^ NK cells as well as potentiate the cytotoxic capacity of CD56^bright^ cells [102]. Therefore, MSC-based delivery of IL2 to tumor sites may increase the number of tumor infiltrating NK cells and increase overall survival of some patients.

In addition to NK cells, increased NKT-leukocyte populations expressing CD3/CD56 were found in PBMCs cultured in hADSCs-IL2 CM. NKT-cells are cytotoxic against tumors, where killing capacity is independent of the T-cell receptor activation [47]. Other studies have also reported that NKT-cell proliferation is IL2-dependent [108,109]. For example, breast cancer cells expressing IL2 were shown to cause the expansion of NKT-cells in vitro [108]. Furthermore, IL2-secreting fibroblasts were shown to increase the NKT-cell counts in patients with peripheral neuroectodermal tumor and, subsequently tumor cell death [109]. High numbers of CD3^+^CD56^+^ NKT cells correlate with the long-term survival of patients with lung and colorectal cancer [110].

In conclusion, although the IL2 genetically modified hADSCs retain MSC properties, they can reduce SH-SY5Y cell proliferation and activate human PBMCs. The IL2 expression-mediated therapeutic effect of hADSCs may be offset by an increased expression of oncogenes, as well as the innate ability of hADSCs to support tumor growth and suppress T-cell proliferation. Therefore, the potential tumor-promoting effects of hADSCs-IL2 requires additional investigation in a range of different cancer contexts before preclinical application.

## Figures and Tables

**Figure 1 bioengineering-07-00059-f001:**
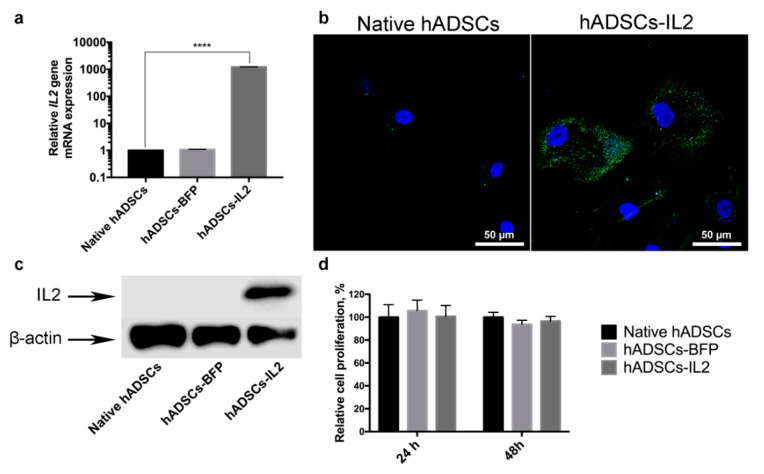
Detection of Interleukin 2 (IL-2) expression in human adipose tissue-derived MSCs were transduced with lentivirus encoding *IL2* (hADSCs-IL2). (**a**) Relative *IL2* gene mRNA expression determined by qPCR was increased by 1000 times in hADSCs-IL2. 18S RNA reference gene has been used for normalization of the data. Bars represent the mean of two biological replicates with their corresponding standard deviation (SD) (*n* = 3). (**b**) Expression of IL2 protein was confirmed using immunofluorescence assay and (**c**) western blot analysis. (**d**) Relative proliferation of hADSCs 24- and 48-h post seeding in vitro was measured using a Cell Proliferation Assay kit (Promega, Madison, WI, USA). Each value is presented as the % in relation to the control (native hADSCs) group. Bars represent the mean ± SD (*n* = 6) of two biological replicates. **** *p* < 0.0001.

**Figure 2 bioengineering-07-00059-f002:**
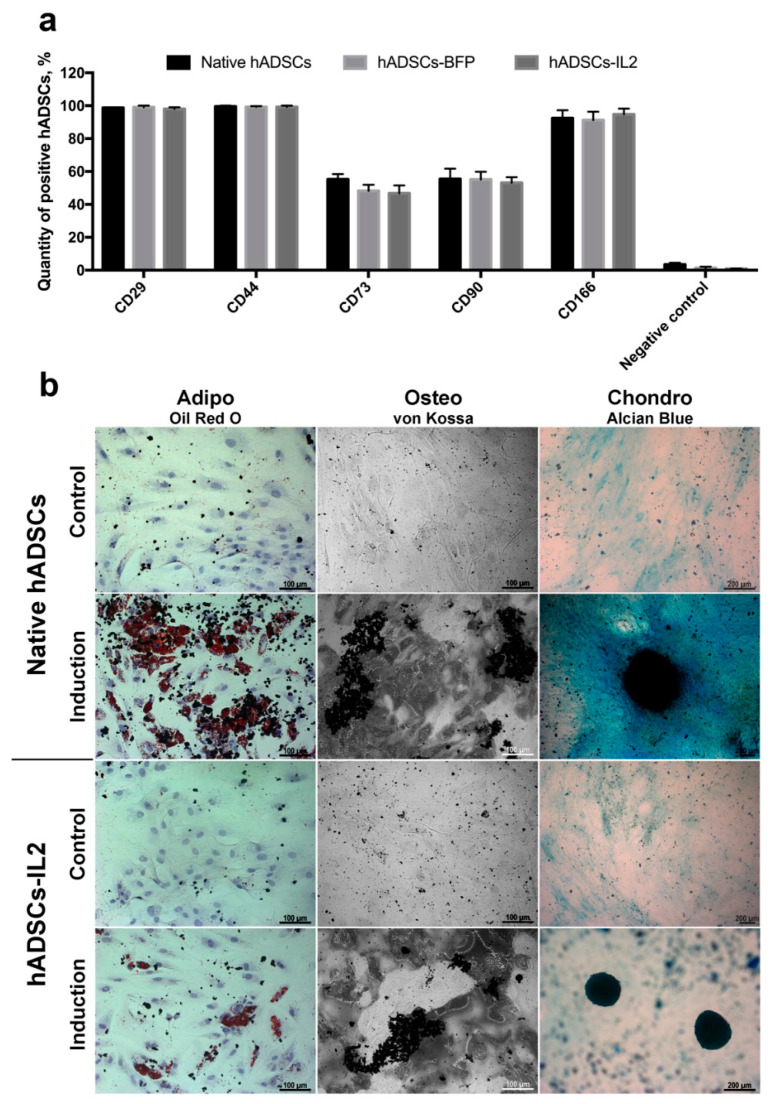
Characterization of the native and genetically modified hADSCs. (**a**) Immunophenotype of native and genetically modified hADSCs. hADSCs were stained with antibodies against specific surface markers CD29, CD44, CD73, CD90, CD166 and negative cocktail (CD11b, CD19, CD34, CD45, and HLA-DR. CD45) and analyzed by flow cytometry. Data are shown as the mean percentage ± SD (*n* = 6) of two biological replicates. (**b**) Osteogenic, adipogenic and chondrogenic differentiation of hADSCs. Phase contrast microscope images. To differentiate toward adipogenic lineage native and IL2-genetically modified cells were cultured in reprogramming medium for 14 days. At day 14, cells were fixed and stained with Oil Red O. For osteogenic differentiation cells were cultured in reprogramming medium for 28 days. At day 28, cells were fixed and analysed by von Kossa staining. Chondrogenic differentiation was determined by staining with Alcian blue on day 21 after seeding (adipogenic and osteogenic differentiation: scale bar, 100 µm; chondrogenic differentiation: scale bar, 200 µm).

**Figure 3 bioengineering-07-00059-f003:**
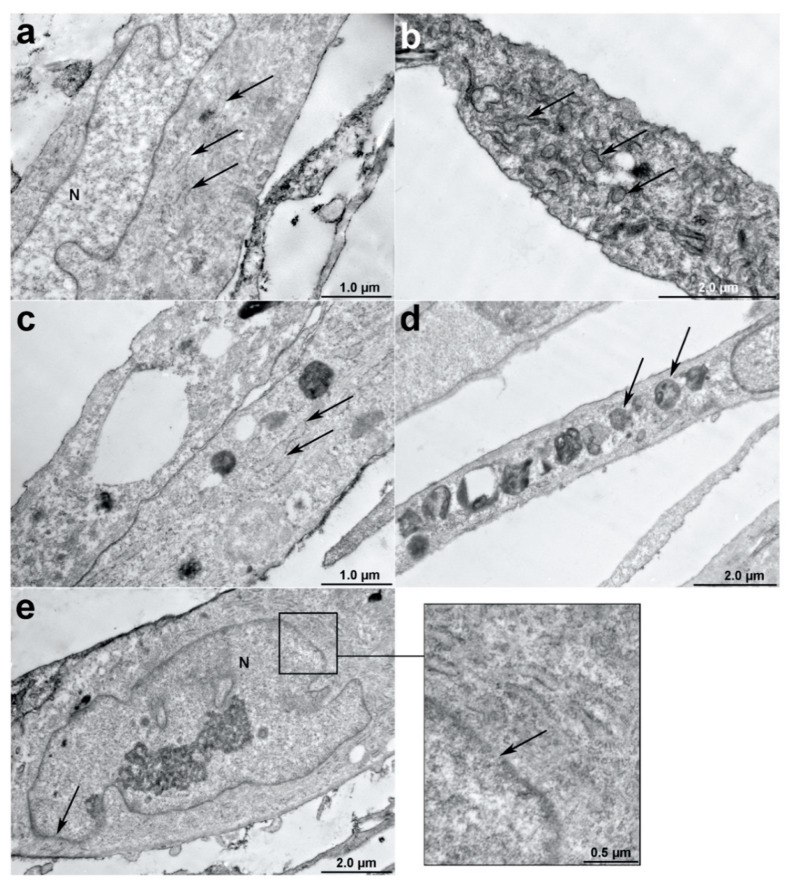
Ultrastructure analysis of native hADSCs, hADSCs-BFP (blue fluorescent protein) and hADSCs-IL2 using transmission electron microscopy (TEM). (**a**) Rough endoplasmic reticulum (ER) of native hADSCs had the form of elongated cisterns with well-distinguishable ribosomes and internal contents of medium electron density (arrows). (**b**) Rough ER of hADSCs-BFP had high electron density (arrows). (**c**) Rough ER of hADSCs-IL2 increased in size and had high electron density (arrows). (**d**) hADSCs-IL2 contain an endosomes (arrows) in the cytoplasm. (**e**) Nuclear pores in hADSCs-IL2 karyolemma (arrow). N—nucleus. Scale bar: 1–2 μm.

**Figure 4 bioengineering-07-00059-f004:**
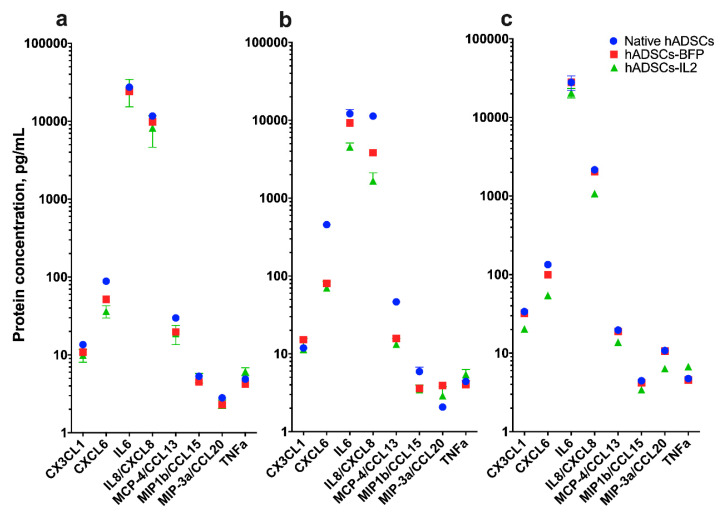
Multiplex analysis of conditioned medium (CM) of native hADSCs, hADSCs-BFP and hADSCs-IL2. CM was harvested from native hADSCs, hADSCs-BFP, hADSCs-IL2 after 24 h of cultivation (**a**), 48 h of cultivation (**b**) and 72 h of cultivation (**c**). Fifty microliters of the sample were used for determining absolute cytokine concentration. Bioplex analysis was performed in two biological replicates, data are shown as the mean percentage ± SD (*n* = 3).

**Figure 5 bioengineering-07-00059-f005:**
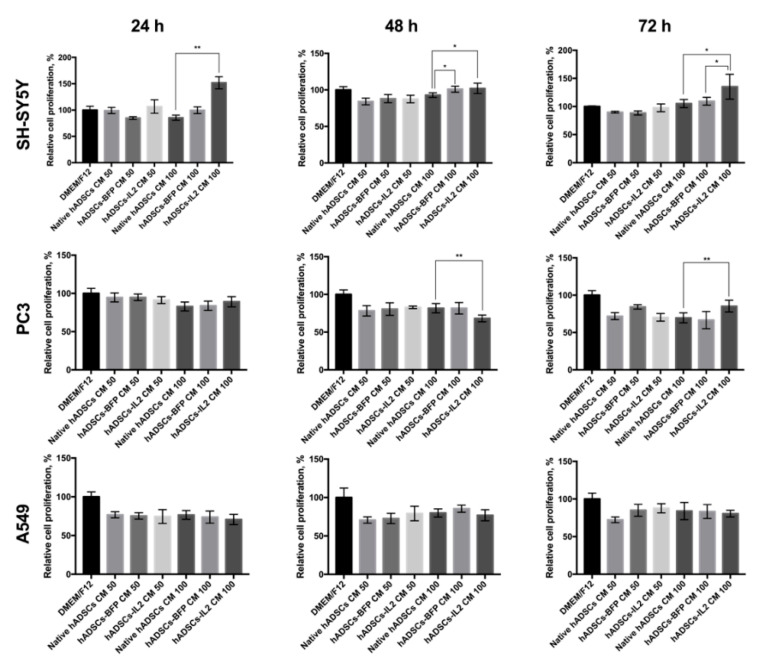
Proliferative activity of SH-SY5Y, PC3 and A549 cancer cells after culturing in CM from hADSCs. SH-SY5Y, A549 or PC3 cells were plated at concentration 5 × 10^3^ cells in 100 μL of DMEM/F12 medium in 96-well plate. Cells were incubated for 24 h (37 °C, 5% CO_2_), the medium was replaced with CM (various dilutions) harvested from the hADSCs at various time points (total volume 100 μL) and incubated for 48 h. Proliferation of hADSCs or cancer cells was evaluated using the CellTiter 96^®^ Aqueous Non-Radioactive Cell Proliferation Assay kit (Promega, Madison, WI, USA). CM 50—medium, containing 50% of Dulbecco’s modified Eagle medium (DMEM)/F12 and 50% of hADSC CM, total volume 100 µL. CM 100—100% hADSC CM, total volume–100 µL. Each box represents the mean ± SD (*n* = 6) of two biological replicates. * *p* < 0.05, ** *p* < 0.01.

**Figure 6 bioengineering-07-00059-f006:**
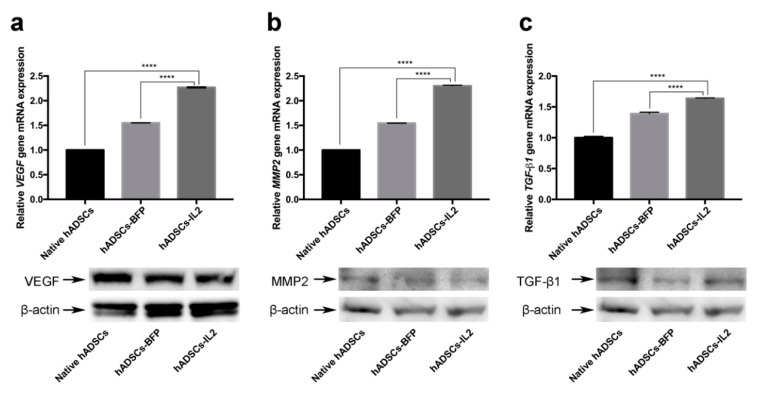
The relative gene expression and protein secretion of vascular endothelial growth factor (VEGF), transforming growth factor β (TGF-β1) and matrix metalloproteinase 2 (MMP2) in native hADSCs, hADSCs-BFP and hADSCs-IL2. (**a**) *VEGF* gene mRNA expression level and VEGF protein analysis. (**b**) *MMP2* gene mRNA expression level and MMP2 protein analysis. (**c**) *TGF-β1* gene mRNA expression level and TGF-β1 protein analysis. 18S RNA reference gene has been used for normalization of qPCR data. Bars represent the mean ± SD (*n* = 3) of two biological replicates. **** *p* < 0.0001.

**Figure 7 bioengineering-07-00059-f007:**
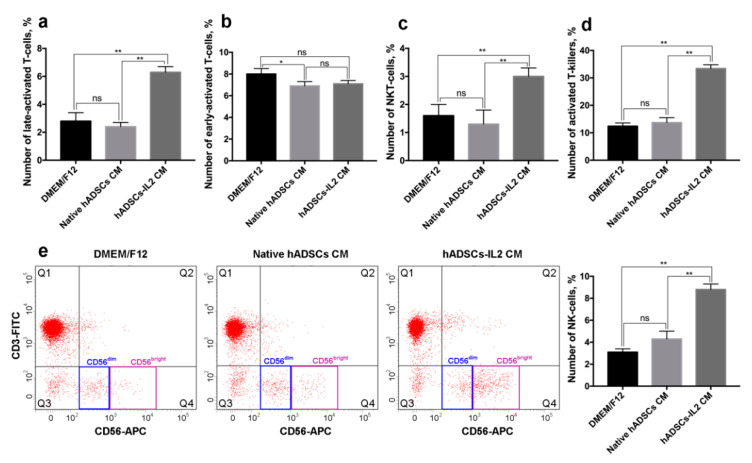
Changes in peripheral blood mononuclear cell (PBMC) activation marker expression after cultivation in hADSCs and hADSCs-IL2 CM. PBMCs were plated in 35 mm non-treated dishes (2 × 10^6^ cells per dish) in 2 mL of CM harvested from hADSCs-IL2, native hADSCs or fresh DMEM/F12. Activation of PBMCs populations was determined using flow cytometry after 72 h. (**a**) Number of CD3^+^/HLA-DR^+^ late-activated T-cells, %. (**b**) Number of CD3^+^/CD25^+^ early-activated T-cells, %. (**c**) Number of CD3^+^/CD56^+^ NKT-cells, %. (**d**) Number of CD8^+^/CD38^+^ activated T-killers, %. (**e**) Dot-plot with CD3^−^/CD56^+^ NK cells cultured in DMEM/F12, hADSC CM, hADSCs-IL2 CM. Each box represents the mean ± SD (*n* = 3) of two biological replicates. * *p* < 0.05, ** *p* < 0.01, ns: no significance.

**Figure 8 bioengineering-07-00059-f008:**
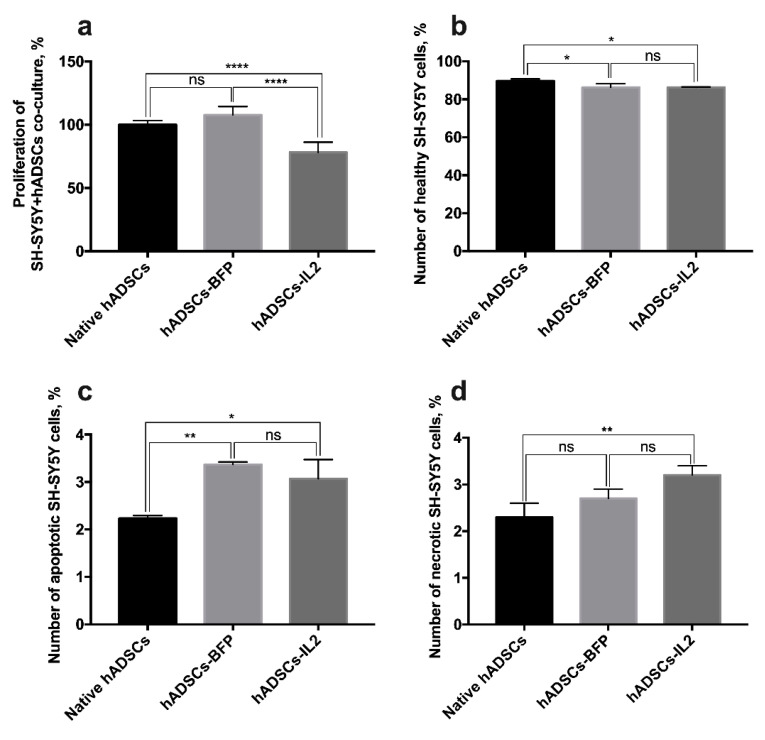
Analysis of SH-SY5Y cell proliferation and viability in co-culture with hADSCs. (**a**) SH-SY5Y were co-cultured with native hADSCs, hADSCs-BFP and hADSCs-IL2 for 72 h and proliferative activity of co-culture was determined using Cell Proliferation Assay kit (Promega, Madison, WI, USA). Each value is presented as the % in relation to the control (native hADSCs) group. Bars represent the mean ± SD (*n* = 6) of two biological replicates. (**b**,**c**) SH-SY5Y-GFP were co-cultured with native hADSCs, hADSCs-BFP and hADSCs-IL2. After 72 h of co-culture, SH-SY5Y-GFP cells were separated from native hADSCs, hADSCs-BFP or hADSCs-IL2 using FACS Aria III (BD Biosciences, San Jose, CA, USA). The apoptotic and necrotic cell counts were determined immediately after separation by Annexin V/PI staining. Number of healthy (**b**), apoptotic (**c**) and necrotic (**d**) SH-SY5Y cells after co-cultivation with hADSCs represents the mean ± SD (*n* = 3) of two biological replicates. * *p* < 0.05, ** *p* < 0.01, **** *p* < 0.0001, ns: no significance.

**Figure 9 bioengineering-07-00059-f009:**
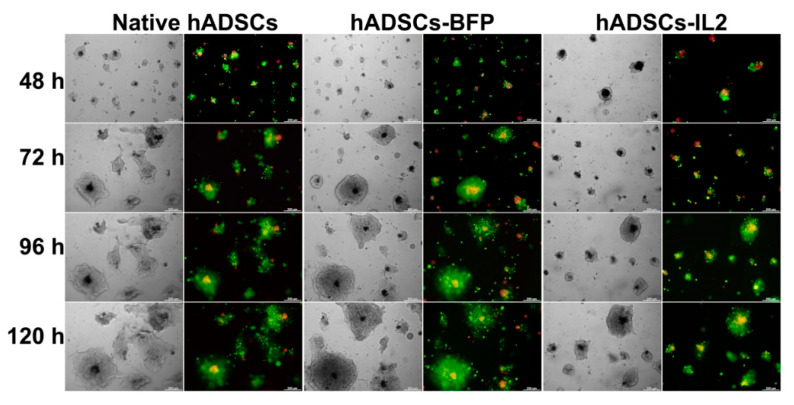
Analysis of the organization of SH-SY5Y and native and genetically modified hADSCs on Matrigel matrix. SH-SY5Y cells and native hADSCs, hADSCs-BFP and hADSCs-IL2 were labelled with DiO (green) and DiD (red) dyes using Vybrant Multicolor Cell-Labeling Kit and were co-cultured for 120 h on a thin layer of Matrigel. Self-organization of cells in co-culture was evaluated after 48, 72, 96 and 120 h using phase-contrast and fluorescence microscopy with inverted Axio Observer.Z1 microscope. After 48 h, the cells organized into individual cellular aggregates with flying saucer-like architecture, in which the core consisted of hADSCs (red) that were surrounded by a flat aureole of SH-SY5Y cells (green). Throughout the observation period, SH-SY5Y cells actively proliferated around stem cells, the proliferation rate of tumor cells was significantly higher than the proliferation rate of hADSCs, which led to the compression of hADSC cultures. In co-culture with hADSCs-IL2 the inhibition of SH-SY5Y tumor cell growth was clearly observed compared to co-cultures of SH-SY5Y cells with native hADSCs or hADSCs-BFP.

**Table 1 bioengineering-07-00059-t001:** Primer and probe sequences of related genes for quantitative polymerase chain reaction (qPCR).

Target Gene	Forward Primer (5′−3′)	Reverse Primer (5′−3′)	TaqMan Probe (5′−3′)
18S rRNA	GCCGCTAGAGGTGAAATTCTTG	CATTCTTGGCAAATGCTTTCG	[HEX] ACCGGCGCAAGACGGACCAG [BH2]
IL2	CACCAGGATGCTCACATTTAAG	GTCCCTGGGTCTTAAGTGAAAG	[FAM] CCCAAGAAGGCCACAGAACTGAAACA [BH1]
VEGF	ATCACCATGCAGATTATGGC	TGCATTCACATTTGTTGTGC	[FAM] TCAAACCTCACCAAGGCCAGCA [BH1]
MMP2	ACCCATTTACACCTACACCAAG	TGTTTGCAGATCTCAGGGTC	[FAM] TCAATGTCAGGAGGCCCCATAGA [BH1]
TGF-β1	GCCTTTCCTGCTTCTCATGG	TCCTTGCGGAAGTCAATGTAC	[FAM] CCGACCCTGGACACCAACTAT [BH1]

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
