# Peer review of "Human Mesenchymal Stem Cells Overexpressing Interleukin 2 Can Suppress Proliferation of Neuroblastoma Cells in Co-Culture and Activate Mononuclear Cells In Vitro"

_bioengineering, 2020, doi:10.3390/bioengineering7020059_

Round 1

Reviewer 1 Report

The manuscript "Human Mesenchymal Stem Cells Overexpressing Interleukin 2 Сan Suppress Proliferation of Neuroblastoma Cells in Co-Culture and Activate  Mononuclear Cells in Vitro" presents an interesting approach to deliver IL2 to the tumor sites with the help of the AD-MSc cells.

The Authors characterized modified cells to parental cells or the one expressing fluorescence protein. They assessed how the conditioned medium from MSC cells or co-culture would affect the growth of neuroblastoma cells in several models.

A question is worth answering in the introduction - if MSC therapy is considered for the neuroblastoma treatment?

A few technical issues interefere with the overall nice flow of the manuscript:

1) The blots in the Fig. 6 - the reader will not see any increases even slight in the blots; MMP2 and TGFb1 antibodies give poor signal, but the actin itself seems to be overexposed. Maybe the Authors have more representative blots? Shorter exposure of actin would do. Increases mentioned by Authors for the protein levels are slight and do not correspond with the mRNA expression but they are not visible in the form presented now.

2) Fig.8b is not convincing and if understood correctly it is from 2 biological experiments. The results should be either presented as maybe necrotic/apopotic fraction in each group or the scale should be adjusted. It is, however, recommended that the third repetition should be included.

Author Response

Thank you for your comments. We have corrected them point by point within the manuscript accordingly (your comments are in bold text and our responses are in ordinary type):

1) The blots in the Fig. 6 - the reader will not see any increases even slight in the blots; MMP2 and TGFb1 antibodies give poor signal, but the actin itself seems to be overexposed. Maybe the Authors have more representative blots? Shorter exposure of actin would do. Increases mentioned by Authors for the protein levels are slight and do not correspond with the mRNA expression but they are not visible in the form presented now.

More representative β-actin bands were added in the Figure 6 for all the proteins. We have also rechecked the results with the new bands. However, no changes were observed compared to previous results.

2) Fig.8b is not convincing and if understood correctly it is from 2 biological experiments. The results should be either presented as maybe necrotic/apopotic fraction in each group or the scale should be adjusted. It is, however, recommended that the third repetition should be included.

We have added the information about the number of necrotic and apoptotic SH-SY5Y cells after the co-culture with hADSCs. Results are described in Page 16, Lines 535-547, Figure 8c-d, and discussed in Page 19, Lines 691-695.

Reviewer 2 Report

Comment:

The manuscript was based on complex literature and attempted to cave out a niche of a crowded field.

“In preclinical and early clinical studies, local application of IL-2 in the tumor is clinical more effective in anticancer therapy than systemic IL-2 therapy, over a broad range of doses, without serious side effects.”  doi:10.1007/s00262-008-0455-z

“Tumour blood vessels are more vulnerable than normal blood vessels to the actions of IL-2. When injected inside a tumor, i.e., local application, a process mechanistically similar to the vascular leakage syndrome, occurs in tumor tissue only. Disruption of the blood flow inside of the tumor effectively destroy tumor tissue.” doi:10.1007/s00262-004-0627-4.

“Co-injection of autologous fibroblasts Transfected With IL-2 and IL-12 with Neuro-2A tumor cells abolished their in vivo tumorigenicity” (doi: 10.1038/sj.bjc.6603857).

“In local application, the systemic dose of IL-2 is too low to cause side effects, since the total dose is about 100 to 1000-fold lower. Clinical studies showed painful injections at the site of radiation as the most important side effect reported by patients. In the case of irradiation of nasopharyngeal carcinoma, the five-year disease-free survival increased from 8% to 63% by local IL-2 therapy” (PMID 15685449). doi: 10.1016/j.canlet.2005.01.057.

Surprisingly in clinical trials, “no evidence that addition of subcutaneous IL-2 to immunotherapy with Immunotherapy with the chimeric anti-GD2 monoclonal antibody dinutuximab beta, given as an 8 h infusion, improved outcomes in patients with high-risk neuroblastoma” (PMID: 30442501 DOI: 10.1016/S1470-2045(18)30578-3). Another report  shows that “A phase 3 randomized study (COG ANBL0032) demonstrated significantly improved outcome by adding immunotherapy with ch14.18 antibody to isotretinoin as post-consolidation therapy for high-risk neuroblastoma (NB), combined with GM-CSF or IL2, however, Of 105 patients enrolled, five patients developed protocol-defined unacceptable toxicities.” (doi: 10.3389/fimmu.2018.01355.)

They, therefore, designed the road maps to determine the effect of hADSCs on activation of peripheral blood mononuclear cells (PBMCs) and proliferation and viability of SH-SY5Y neuroblastoma cells after co-culture with native hADSCs, hADSCs-BFP or hADSCs-IL2 on plastic and Matrigel matrices. They speculated on the mechanisms by which the in vitro data might help the related therapies. It is of great interest to Neuroblastoma, as its treatment is still dismal, and any approach is appreciated. However, they might improve their clarity and logic flow by addressing the following 17 specific comments.

Specific comments:

  • Lines 33-36: “Conditioned medium from hADSC-IL2 affected tumor cell proliferation, increasing the proliferation of SH-SY5Y cells and also increasing the number of late-activated T-cells, NK cells, NKT-cells and activated T-killers. Conversely, the hADSC-IL2 co-culture led to a decrease in SH-SY5Y 35 proliferation on plastic and Matrigel.” What was the underlined mechanism for such opposite effects? What was the ratio among increasing the number of late-activated T-cells, NK cells, NKT-cells, and activated T-killers?
  • Lines 23-24: “However, systemic administration of high doses of IL2 can be toxic, causing capillary leakage syndrome and stimulating the pro-tumor immune response.” Any adverse effects of their hADSC-IL2 approach? They did not have data to support such an introduction. Lines 537-545 touched it but not sufficient with only in vitro data for NB tumor type (Lines 577-614), mostly based on loose speculation of other tumor types.
  • Lines 36-37: “These data show that hADSCs-IL2 can reduce SH-SY5Y proliferation and activate PBMCs.” The authors should specify this is only valid for in vitro.
  • What did they assay to quality control of hADSC-IL2 remain stem cell characters? Their supporting data? E.g., did hADSC-IL2 affect differentiation?
  • Lines 514-516: “Throughout the observation period, SH-SY5Y cells actively proliferated around stem cells, the proliferation rate of tumor cells was significantly higher than the proliferation rate of hADSCs, which led to the compression of hADSC cultures.” Where was the data for the claim? The ratio? The number of both cells? How did they track both? It was not clear from Fig. 8 and Fig. 9. They should have put all the controls: with native hADSCs alone, hADSCs-BFP alone, hADSCs-IL2 alone, SH-SY5Y cells alone, side-by-side with co-culture. They should have put all panels up for comparison, both BFP blue (ADSC) and GFP green SH-SY5Y, with co-labeling of perspective biomarkers.  It is of confusing that “SH-SY5Y cells and native hADSCs, hADSCs-BFP and hADSCs-IL2 were labeled with DiO 509 (green) and DiD (red) dyes using Vybrant Multicolor Cell-Labeling Kit and were co-cultured for 120 hours on a 510 thin layer of Matrigel.” (Lines 509-511).
  • Lines 398-404: “398 tumor growth and metastasis, as well as stimulate neovascularization by expressing multiple pro-399 angiogenic and trophic factors such as VEGF, IL8, TGF-β, EGF, and PDGF [15]. In this context, 400 increased pro-oncogene expression in response to further modifications of MSCs could cause a significant potential problem when using modified MSCs for patient treatment. Therefore, we sought to determine the effect of IL2 overexpression on the abundance of VEGF, MMP2, and TGF-β1 genes, known to have pro-oncogenic and pro-angiogenic properties [40-42] in hADSCs-IL2.” They used this to set up for Fig. 7; however, they omitted the data that show the suppression of tumor growth by MSCs, as they did talk in the introduction. Thus, both Fig. 6 and Fig. 7 were not quite balanced out in the logic of why they found that here.
  • 6. How long did they collect the conditional medium? Time-point? Time course? How did they pick up the time?
  • 5: “CM volume was total volume 100 μl” – how many cells? Petri dish size? “CM 50 — medium, containing 50% of DMEM/F12 and 50% of hADSC CM.” total volume of culture media? “Each box represents the mean ± SD (n = 6) of two biological replicates” Can they define “replicates” and “n=6” in the context? Why is Fig. 4, n=3?
  • 4: Why did they not show IL2 level among hADSCs, hADSCs-BFP, hADSCs-IL2 cells?
  • 3. Why did they not use the same scales for all cells? How many view fields did they pick up for “Rough ER of hADSCs-IL2 increased in size and had high electron density (arrows)?” Given the heterogeneity of ADSCs, I would think you could find whatever you look for under TEM.
  • 2. “percentage ± SD (n = 3) of two biological replicates.” Can they define how they pick up such two biological replicates? FACS got 100% of CD29, CD44, CD166? Could they come up with literature support for the numbers? Negative control was not a reasonable control. Lines 292-294: “Immunofluorescence analysis of native hADSCs, hADSCs-BFP, and hADSCs-IL2 determined that the majority of native and genetically modified hADSCs shared the same pattern of surface antigen expression, which was similar to that commonly detected on human MSCs.” If “that the majority of native and genetically modified hADSCs shared the same pattern of surface antigen expression, which was similar to that commonly detected on human MSCs, “ how did they gain 100% FACS markers?

Lines 105 – 106: “hADSCs were used up to 105 passages 5-7” – It has been well known that such passage numbers, the cells lose its biomarker expression.

  • 1. “(b) Expression of IL2 protein was confirmed using immunofluorescence assay.” That alone could not confirm protein expression, as the size of the recombinant protein was unknown. What were the time points for panels a & b? Given panel c, no differences were found among three lines. How did they explain the differences in Panels a & b?
  • Line 94: “Adipose tissue and blood samples were collected from donors at” – where was the blood data? Why did they not add it to those figures?
  • Abstract: “However, IL2-mediated therapeutic effects of hADSCs could be offset by the increased expression of pro-oncogenes, as well as the natural ability of hADSCs to promote the progression of some tumors.” This sentence was not tied to their data.
  • Lines 89-91: Their data do not accurately support the statement.
  • Lines 614-650: Neither did their data support nor was relevant with in vitro data as reported. Some citations need for their elaboration.
  • Lines 525-529: “However, overexpression of IL2 did cause significant changes in hADSCs-IL2 ultrastructure, where an increased size and density of rough ER and the Golgi apparatus were noted. However, significant changes in the ultrastructure of GC and ER in hADSCs-BFP were also seen when compared to native hADSCs. Therefore, these changes may be linked to recombinant protein overproduction rather than IL2 directly.” It was not convincing without immune-gold EM confirmation of the IL2 protein expression. Alternatively, confocal immune-microscopy of co-labeling would have helped out to the conclusion.

Author Response

Thank you for your comments which have helped us to improve our manuscript. The information provided was very useful so we have broadened the Introduction (Page 2, Lines 62-66) and Discussion sections (Page 19, Lines 720-729). We have also corrected your comments point by point within the manuscript accordingly (your comments are in bold text and our responses are in ordinary type):

1) Lines 33-36: “Conditioned medium from hADSC-IL2 affected tumor cell proliferation, increasing the proliferation of SH-SY5Y cells and also increasing the number of late-activated T-cells, NK cells, NKT-cells and activated T-killers. Conversely, the hADSC-IL2 co-culture led to a decrease in SH-SY5Y 35 proliferation on plastic and Matrigel.” What was the underlined mechanism for such opposite effects? What was the ratio among increasing the number of late-activated T-cells, NK cells, NKT-cells, and activated T-killers?

Possible mechanisms for the opposite effects of CM and co-culture on the proliferation of SH-SY5Y tumor cells have been discussed in Page 20, Lines 729-734. The diverse effects of CM harvested from hADSCs-IL2 versus direct co-culture with hADSCs-IL2 on proliferation of SH-SY5Y could be explained by the level of biologically active molecules in CM generated by a large number of hADSCs-IL2 cells (compared to the number of hADSCs-IL2 cells in co-culture) or by direct cell-cell inhibitory effects of hADSCs-IL2 due to changes in surface protein expression or other cellular factors altered by IL2 overexpression on SH-SY5Y in co-culture.

The number of late-activated T-cells, NK cells, NKT-cells, and activated T-killers was increased in 3 times, 4 times, 2 times and 3 times respectively (described in section 3.6 hADSCs-IL2 activates mononuclear blood cells in vitro; Page 14, Lines 489-512). Due to the word limit for Abstract section we confined ourselves to general description of the results.

2) Lines 23-24: “However, systemic administration of high doses of IL2 can be toxic, causing capillary leakage syndrome and stimulating the pro-tumor immune response.” Any adverse effects of their hADSC-IL2 approach? They did not have data to support such an introduction. Lines 537-545 touched it but not sufficient with only in vitro data for NB tumor type (Lines 577-614), mostly based on loose speculation of other tumor types.

Systemic toxicity of high doses of IL2 has been described in patients in clinical trials. Regarding the toxicity of hADSC-IL2, it will be evaluated in further studies in animal models.

3) Lines 36-37: “These data show that hADSCs-IL2 can reduce SH-SY5Y proliferation and activate PBMCs.” The authors should specify this is only valid for in vitro.

It was clarified.

4) What did they assay to quality control of hADSC-IL2 remain stem cell characters? Their supporting data? E.g., did hADSC-IL2 affect differentiation?

To improve the assessment of stem cell characteristics of genetically modified hADSCs, we added the results of the differentiation of native hADSCs and hADSCs-IL2 into adipogenic, chondrogenic and osteogenic directions (Page 3, Lines 126-144; Page 8, Lines 337-340; Figure 2b).

5) Lines 514-516: “Throughout the observation period, SH-SY5Y cells actively proliferated around stem cells, the proliferation rate of tumor cells was significantly higher than the proliferation rate of hADSCs, which led to the compression of hADSC cultures.” Where was the data for the claim? The ratio? The number of both cells? How did they track both? It was not clear from Fig. 8 and Fig. 9. They should have put all the controls: with native hADSCs alone, hADSCs-BFP alone, hADSCs-IL2 alone, SH-SY5Y cells alone, side-by-side with co-culture. They should have put all panels up for comparison, both BFP blue (ADSC) and GFP green SH-SY5Y, with co-labeling of perspective biomarkers.  It is of confusing that “SH-SY5Y cells and native hADSCs, hADSCs-BFP and hADSCs-IL2 were labeled with DiO 509 (green) and DiD (red) dyes using Vybrant Multicolor Cell-Labeling Kit and were co-cultured for 120 hours on a 510 thin layer of Matrigel.” (Lines 509-511).

For co-cultivation, native and genetically modified hADSCs and native SH-SY5Y (5×103 cells of each cell type per well in 1 mL of DMEM/F12) were used. SH-SY5Y-GFP were used only for the apoptosis/necrosis test. Therefore, in the experiment with co-cultures, SH-SY5Y cells were stained with DiO (green), and hADSCs were stained with DiD (red), including hADSCs-BFP, because BFP rapidly loses its fluorescence during microscopy. It has been previously shown that hADSCs form capillary-like structures on Matrigel (PMID: 23056481), while SH-SY5Y cells represent the morphology they have when cultured on plastic (PMID: 22367235). Therefore, as a control for assessing the interaction of MSCs and tumor cells, we used co-cultures on plastic, the data are not presented since no significant difference between the cultures was observed, and we believe that this information would be redundant. The appropriate information was added in the manuscript (Page 19, Lines 701-703).

6) Lines 398-404: “398 tumor growth and metastasis, as well as stimulate neovascularization by expressing multiple pro-399 angiogenic and trophic factors such as VEGF, IL8, TGF-β, EGF, and PDGF [15]. In this context, 400 increased pro-oncogene expression in response to further modifications of MSCs could cause a significant potential problem when using modified MSCs for patient treatment. Therefore, we sought to determine the effect of IL2 overexpression on the abundance of VEGF, MMP2, and TGF-β1 genes, known to have pro-oncogenic and pro-angiogenic properties [40-42] in hADSCs-IL2.” They used this to set up for Fig. 7; however, they omitted the data that show the suppression of tumor growth by MSCs, as they did talk in the introduction. Thus, both Fig. 6 and Fig. 7 were not quite balanced out in the logic of why they found that here.

It was clarified in the manuscript that MSCs have anti- and pro-tumor properties (Page 13, Line 442-443). Subsection 3.5 hADSCs-IL2 can promote angiogenesis and increase tumor cell invasion discusses the possible pro-tumor effects of MSCs, which may be enhanced due to genetic modification. The ability of MSCs to suppress SH-SY5Y proliferation is discussed in subsection 3.7.

7) How long did they collect the conditional medium? Time-point? Time course? How did they pick up the time?

Collection of conditioned medium is described in Materials and Methods section (Page 6, Lines 229-232). Native hADSCs, hADSCs-BFP and hADSCs-IL2 were seeded at a density of 2 × 105 cells in T75 culture flasks. Conditioned medium (CM) was harvested after 24, 48 and 72 hours of cultivation to analyze the changes in protein profile at various time points and how it influence at tumor and immune cells.

8) “CM volume was total volume 100 μl” – how many cells? Petri dish size? “CM 50 — medium, containing 50% of DMEM/F12 and 50% of hADSC CM.” total volume of culture media? “Each box represents the mean ± SD (n = 6) of two biological replicates” Can they define “replicates” and “n=6” in the context? Why is Fig. 4, n=3?

The information about total medium volumes was added in the manuscript. The experiment was carried out in six technical and two biological replicates. The experiment in Figure 4 was carried out in three technical replicates. 

9) Why did they not show IL2 level among hADSCs, hADSCs-BFP, hADSCs-IL2 cells?

Western blot analysis of IL2 production in native hADSCs, hADSCs-BFP, hADSCs-IL2 was added in the manuscript (Page 7, Line 321; Figure 1c).

10) Why did they not use the same scales for all cells? How many view fields did they pick up for “Rough ER of hADSCs-IL2 increased in size and had high electron density (arrows)?” Given the heterogeneity of ADSCs, I would think you could find whatever you look for under TEM.

Various magnification (1-0.5 µm scale) was used to find out changes in organelle structures, since one structures are better seen at a low magnification, and others are at a high magnification. Three grids were examined for each sample, 15 view fields were examined on each grid.

11) “percentage ± SD (n = 3) of two biological replicates.” Can they define how they pick up such two biological replicates? FACS got 100% of CD29, CD44, CD166? Could they come up with literature support for the numbers? Negative control was not a reasonable control. Lines 292-294: “Immunofluorescence analysis of native hADSCs, hADSCs-BFP, and hADSCs-IL2 determined that the majority of native and genetically modified hADSCs shared the same pattern of surface antigen expression, which was similar to that commonly detected on human MSCs.” If “that the majority of native and genetically modified hADSCs shared the same pattern of surface antigen expression, which was similar to that commonly detected on human MSCs, “ how did they gain 100% FACS markers? Lines 105 – 106: “hADSCs were used up to 105 passages 5-7” – It has been well known that such passage numbers, the cells lose its biomarker expression.

To evaluate two biological replicates two different hADSCs from different donors were used to carry out the experiments. Standard immunophenotyping (BD StemflowTM Human MSC Analysis Kit, BD Biosciences, USA) of newly isolated hADSCs include negative control to exclude the presence of hematopoietic stem cells, which may remain after the isolation from adipose tissue. And hADCSs at low passages express lots of CD29 and CD44 markers (DOI: 10.2147/IJN.S244453, PMID: 25924984). The immunopenotyping was carried at low passages so most of the cells expressed typical MSC markers.

12) 1 (b) Expression of IL2 protein was confirmed using immunofluorescence assay.” That alone could not confirm protein expression, as the size of the recombinant protein was unknown. What were the time points for panels a & b? Given panel c, no differences were found among three lines. How did they explain the differences in Panels a & b?

To confirm IL2 protein production western blot analysis of native hADSCs, hADSCs-BFP, hADSCs-IL2 was added in the manuscript (Page 7, Line 321; Figure 1c). qPCR and immunofluorescent analysis were carried out after the Blasticidin selection. In panel C there is no changes in the proliferation rate of hADSCs which means that genetic modification fails to affect hADSC proliferation rate.

13) Line 94: “Adipose tissue and blood samples were collected from donors at” – where was the blood data? Why did they not add it to those figures?

Blood samples were collected from healthy donors and isolated as described in Materials and Methods Section (Page 3, Lines 114-117).

14) Abstract: “However, IL2-mediated therapeutic effects of hADSCs could be offset by the increased expression of pro-oncogenes, as well as the natural ability of hADSCs to promote the progression of some tumors.” This sentence was not tied to their data.

It was corrected.

15) Lines 89-91: Their data do not accurately support the statement.

The statement was rephrased to avoid misunderstanding.

16) Lines 614-650: Neither did their data support nor was relevant with in vitro data as reported. Some citations need for their elaboration.

We added the information about some investigation to improve the discussion of our results (Page 19, Lines 677-679; Page 19, Lines 691-695).

17) Lines 525-529: “However, overexpression of IL2 did cause significant changes in hADSCs-IL2 ultrastructure, where an increased size and density of rough ER and the Golgi apparatus were noted. However, significant changes in the ultrastructure of GC and ER in hADSCs-BFP were also seen when compared to native hADSCs. Therefore, these changes may be linked to recombinant protein overproduction rather than IL2 directly.” It was not convincing without immune-gold EM confirmation of the IL2 protein expression. Alternatively, confocal immune-microscopy of co-labeling would have helped out to the conclusion.

We do not claim that increased size and density of rough ER is caused by IL2 overexpression, however, the changes in shape, an increase in the area and electron density of ER are signs of an increase in the protein expression on the ultrastructural level (discussed in Page 17, Lines 584-591).

Round 2

Reviewer 1 Report

All of the comments have been addressed properly.

Reviewer 2 Report

accepted